# SensorLM: Learning the Language of Wearable Sensors

Yuwei Zhang[1,3,†,*]  Kumar Ayush[1,*]  Siyuan Qiao[2]  A. Ali Heydari[1]  Girish Narayanswamy[†]
Maxwell A. Xu[†]  Ahmed A. Metwally[1]  Shawn Xu[1]  Jake Garrison[1]  Xuhai Xu[1]  Tim Althoff[1]
Yun Liu[1]  Pushmeet Kohli[2]  Jiening Zhan[1]  Mark Malhotra[1]  Shwetak Patel[1]
Cecilia Mascolo[3]  Xin Liu[1]  Daniel McDuff[1]  Yuzhe Yang[1,4,†]

[1]Google Research   [2]Google DeepMind   [3]University of Cambridge
[4]University of California, Los Angeles

## Abstract

We present `SensorLM`, a family of sensor-language foundation models that enable wearable sensor data understanding with natural language. Despite its pervasive nature, aligning and interpreting sensor data with language remains challenging due to the lack of paired, richly annotated sensor-text descriptions in uncurated, real-world wearable data. We introduce a hierarchical caption generation pipeline designed to capture *statistical*, *structural*, and *semantic* information from sensor data. This approach enabled the curation of the largest sensor-language dataset to date, comprising over 59.7 million hours of data from more than 103,000 people. Furthermore, `SensorLM` extends prominent multimodal pretraining architectures (e.g., CLIP, CoCa) and recovers them as specific variants within a generic architecture. Extensive experiments on real-world tasks in human activity analysis and healthcare verify the superior performance of `SensorLM` over state-of-the-art in zero-shot recognition, few-shot learning, and cross-modal retrieval. `SensorLM` also demonstrates intriguing capabilities including scaling behaviors, label efficiency, sensor captioning, and zero-shot generalization to unseen tasks. Code is available at https://github.com/Google-Health/consumer-health-research/tree/main/sensorlm.

## 1 Introduction

The human experience unfolds as a continuous dialogue between ***sensory perception*** and ***language articulation***. A perceived change in raw physiological readings like *"heart rate"*, can be seamlessly translated from low-level statistical descriptions (e.g., *"heart rate spikes from 65 to 90"*) to high-level semantic abstractions (e.g., *"periods of strength training noted"*). This intrinsic interplay between raw sensation and linguistic abstraction forms human comprehension of internal states, health conditions, behavioral changes, and more [7, 16]. Wearable sensors now record these narratives at unprecedented resolution, producing *minute-level* data that captures dense insights into an individual's physiological and behavioral states [24, 41]. Aligning and interpreting this rich, continuous stream of fine-grained sensor data with intuitive and actionable language descriptions is critical for user engagement [40], clinical decision-making [38], personalized insights [7], and behavioral interventions [26].

However, directly interpreting continuous raw sensor data poses significant challenges. While large language models (LLMs) excel at processing textual sequences, they inherently struggle with the high-dimensional, continuous, and temporally extensive nature of wearable data [21]. For example, a single day of minute-level multimodal sensor recordings (see Fig. 1(a)) would expand to over 200,000 tokens [24], far exceeding the practical context length limits of most contemporary LLMs [30, 32]. Interestingly, even with minimal subsampling to fit within context constraints, LLMs fail to perform

---

*Co-first authors. †Work done while at Google Research.

39th Conference on Neural Information Processing Systems (NeurIPS 2025).

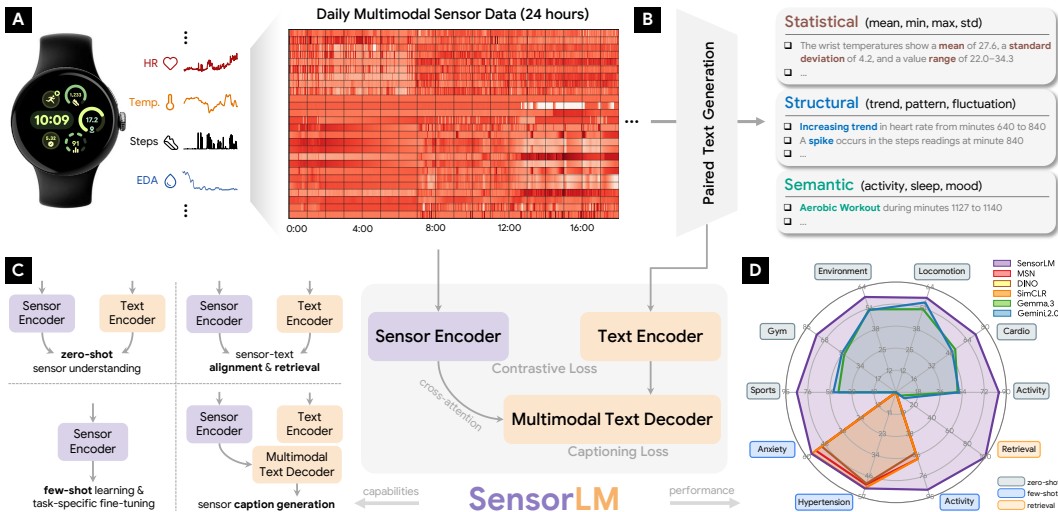

Figure 1: **Sensor-language foundation models (`SensorLM`) for wearable data.** Aligning and interpreting sensor data with natural language remains challenging. **(A)** We present a comprehensive study using over 59.7 million hours of multimodal wearable data from over 103,000 individuals. **(B)** We design a hierarchical pipeline for automatic paired text generation that covers statistical, structural, and semantic sensor information. **(C)** The `SensorLM` pretraining framework and its use cases for diverse downstream tasks. **(D)** Radar plot comparing the performance of `SensorLM` and baselines across various tasks and settings (details in Sec. 5).

well: Fig. 1(d) illustrates the poor zero-shot performance of Gemma-3-27B [30] and Gemini 2.0 [29] (details in Sec. 5). As such, standard LLM methods either become overwhelmed by data volume or sacrifice the granularity needed for aligning sensor events with meaningful linguistic descriptions.

Furthermore, despite preliminary efforts in constructing paired sensor-text datasets to facilitate effective cross-modal alignment, the absence of *large-scale*, *high-quality* sensor-language corpora remains a significant bottleneck [6, 10, 15, 40]. As illustrated in Table 1, current sensor-text datasets are *significantly* limited in scale and primarily structured in restrictive question-answering formats [6, 15, 40]. This highlights a fundamental gap – the lack of principled and scalable methods for generating truly large-scale and diverse sensor-language datasets. Consequently, existing approaches yield sparse and narrowly scoped sensor-text pairs, insufficient for training generalizable foundation models capable of comprehensive multimodal understanding across diverse, open-ended tasks.

To fill the gap, we introduce `SensorLM`, a family of sensor-language foundation models that unlock meaningful interpretation of raw wearable data and enable novel sensor applications through natural language. The effectiveness of `SensorLM` stems from three innovations: (1) a hierarchical, automated caption-generation pipeline that systematically captures multi-level features–*statistical*, *structural*, and *semantic*–from fine-grained sensor streams; (2) building upon this pipeline, the curation of the largest-scale sensor-language dataset to date, comprising over 59 million hours of wearable data from more than 103,000 individuals; and (3) a generic pretraining framework that integrates diverse multimodal architectures (e.g., CLIP [25], Cap [33], CoCa [39]) for scalable and robust learning.

To rigorously evaluate `SensorLM`, we benchmark its performance against state-of-the-art (SOTA) methods across a diverse range of real-world tasks in domains such as human activity analysis and metabolic health. Extensive experiments verify the efficacy of `SensorLM` on multimodal understanding and its ability to enable new sensor-driven applications. Our contributions are as follows:

- We introduce a hierarchical captioning pipeline for raw sensor data, enabling the curation of the largest sensor-language study to date with over 59 million hours of data from over 103,000 people.
- We design `SensorLM`, a family of sensor-language foundation models that enable diverse sensor capabilities through natural language.
- We conduct extensive experiments across various tasks in *human activity analysis* and *metabolic health*, verifying the superior performance of `SensorLM` against SOTA methods.
- Further analyses reveal intriguing properties of `SensorLM` on its scaling behaviors, data efficiency, multimodal understanding & generation, and zero-shot generalization to unseen tasks and concepts.

Table 1: **Comparisons of studies on sensor-text modeling.**

| Study | # People | # Hours (000s) | Sensors[†] | | | | | Text | | |
|---|---|---|---|---|---|---|---|---|---|---|
| | | | PPG | ACC | EDA | TEMP | ALT | statistical | structural | semantic |
| Xing *et al.* [36] | 4 | ≪1 | ✗ | ✓ | ✗ | ✗ | ✗ | ✗ | ✗ | ✓ |
| Moon *et al.* [23] | 60 | <1 | ✗ | ✓ | ✗ | ✗ | ✗ | ✗ | ✗ | ✓ |
| Yu *et al.* [40] | 60 | 5 | ✗ | ✓ | ✗ | ✗ | ✗ | ✗ | ✗ | ✓ |
| Li *et al.* [15] | 214 | 3.7 | ✗ | ✓ | ✗ | ✗ | ✗ | ✗ | ✓ | ✓ |
| SensorLM (ours) | 103,731 | 59,749 | ✓ | ✓ | ✓ | ✓ | ✓ | ✓ | ✓ | ✓ |

[†] PPG: Photoplethysmography.  ACC: Accelerometer.  EDA: Electrodermal Activity.  TEMP: Temperature.  ALT: Altitude.

## 2 Related Work

**Aligning Sensor with Language.** Despite growing interests in integrating sensor data with natural language, large-scale paired corpora derived from uncurated wearable data remain scarce, yet are crucial for training effective cross-modal models [25, 35, 39]. Prior work typically addresses this integration by using pre-derived textual summaries of sensor features or ingesting raw sensor values as tabular inputs to LLMs in downstream prediction tasks [7, 13, 16, 20]. Other multimodal methods incorporate specialized sensor encoders with alignment modules to enhance an LLM's capability to interpret sensor data [6, 10, 14]. Recent efforts initiate the creation of sensor-text data primarily through employing sensor question-answering frameworks [15, 40], explicitly tailored towards human activity recognition. However, these approaches often rely on manual summarization and are inherently limited by sparse and narrowly-scoped sensor-text pairs, restricting generalization to broader scenarios (see Table 1). In contrast, we focus on learning *aligned* sensor-language representations directly from *large-scale*, *hierarchical* captions, enabling strong zero-shot generalization across diverse tasks.

**Multimodal Sensor Foundation Models.** Recent advances in sensor data modeling have demonstrated improved accuracy, robustness, and generalization by leveraging self-supervised pretraining on large-scale physiological and wearable sensor datasets [24, 41]. Existing sensor foundation models primarily focus on single-modal but multi-channel sensor data, utilizing either contrastive-based [1, 31, 37, 41] or reconstruction-based pretraining objectives [24]. More recent efforts have explored multimodal sensor foundation models by aligning individual physiological signals (e.g., ECG, EEG) or IMU sensor data with other modalities, including language or vision, through joint representation learning [11, 23, 45]. Our work extends sensor foundation models towards comprehensive *multimodal sensor-language* understanding, enabling novel sensor applications (see Fig. 1) and *multimodal* capabilities via joint modeling of diverse sensor types and natural language.

**Vision-Language Pretraining.** Vision-language pretraining (VLP) has significantly advanced multimodal AI, enabling models to effectively learn representations that integrate visual and textual data [25, 39]. Leveraging large-scale image-text datasets, VLP methods have produced multimodal foundation models capable of strong zero-shot image classification, captioning, visual question answering, and cross-modal retrieval [25, 35, 39]. Key VLP methods include contrastive learning (e.g., CLIP [25]), which aligns image and text embeddings while separating dissimilar pairs, and generative pretraining (e.g., SimVLM [35]), which employs a prefix language modeling objective. More recent models like CoCa [39] combine both contrastive and generative objectives for enhanced multimodal performance. The success of VLP has inspired similar pretraining strategies in other modalities such as video, audio, time series [8, 23, 44], as well as specialized domains such as medicine [9, 19, 46]. Our work extends VLP paradigms to the *sensor domain*, implementing a generic pretraining framework that incorporates prominent multimodal architectures for scalable sensor-language modeling.

## 3 Sensor-Language Dataset Construction

Building a sensor-language foundation model requires two essential components: (1) a large-scale, diverse collection of *sensor data*, and (2) corresponding *captions* that capture useful properties of the data. We summarize prior efforts on sensor-text modeling in Table 1, and detail our study below.

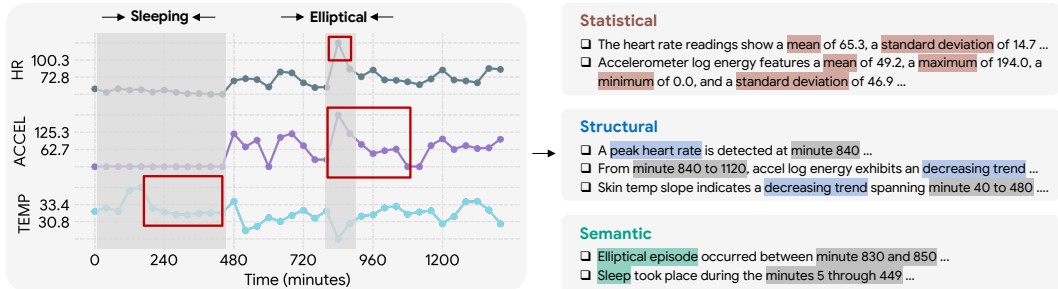

Figure 2: **Hierarchical sensor caption generation pipeline.** Given multimodal wearable data, we generate three levels of captions: (1) *statistical*, capturing basic quantitative metrics; (2) *structural*, describing temporal patterns, trends, and signal dynamics; and (3) *semantic*, providing contextual information such as behavioral states or physical activity. When applicable, each caption is annotated with its time frame (marked in grey).

## 3.1 Sensor Data and Processing

We collected wrist-worn wearable sensor data from Fitbit and Pixel Watch devices. The dataset includes multimodal time series signals from photoplethysmography (PPG), 3-axis accelerometer (ACC), altimeter (ALT), skin temperature (TEMP), and electrodermal activity (EDA) sensors. From these time series, we extracted 26 features: twelve metrics related to heart rate and heart rate variability (HRV) derived from the PPG signal, ten accelerometer-based features (e.g., jerk, steps), as well as the mean and slope of skin temperature, mean tonic electrodermal activity, and mean altimeter pressure (full details are in Appendix A.1). The input to the model consists of one-day windows of sensor data at minutely resolution, resulting in input matrices of shape [26 features × 1440 minutes].

## 3.2 Hierarchical Sensor Caption Generation

A significant challenge in sensor-language pretraining is the general absence of paired textual descriptions for wearable sensor data. Unlike domains such as vision or clinical imaging, where data often comes with associated text (e.g., image captions or clinical reports) [9, 25, 39], sensor data collected in the wild typically lacks such naturally occurring pairings. Existing annotations, when available, are often sparse and limited to coarse labels such as discrete activity categories (Table 1).

To bridge this gap and enable robust alignment between natural language and multimodal sensor streams, we propose a ***hierarchical*** caption generation strategy. This approach creates captions at *three* distinct levels of abstraction—statistical, structural, and semantic. By doing so, we aim to train a model capable of representing not only basic numerical summaries and dynamic temporal patterns, but also high-level events and contextual states observed in daily sensor data. An illustrative example of the overall process and all three level of captions is shown in Fig. 2.

**Statistical Captions.** Statistical captions provide a quantitative summary of the sensor data. For each distinct sensor feature channel (e.g., heart rate, temperature, step count), basic statistical measures–including *mean*, *maximum*, *minimum*, and *standard deviation*–are computed. To enhance linguistic diversity, these values are embedded into a variety of sentence structures using a set of templates (details in Appendix A.2). This offers a concise, data-driven overview of individual sensor streams.

**Structural Captions.** Structural captions focus on encoding the dynamic characteristics and patterns within time-series sensor data. This includes identifying and describing notable *trends*, *fluctuations*, and other *temporal features* across sensor channels. Methodologically, we apply sliding windows to each time series to detect significant increasing, decreasing, or stable trends, as well as identifying sharp spikes and drops. Similar to statistical captions, these identified patterns are then verbalized using a diverse set of templates, from which a random subset is selected for each data sample. This enables the model to learn the temporal shape and behavioral dynamics of the sensor signals.

**Semantic Captions.** Semantic captions aim to capture the high-level meaning and context embedded in the sensor data, reflecting what the individual might be doing or experiencing. This layer incorporates information about recognized activities and sleep periods, specifying start and end times of each event–e.g., *"Observed Outdoor Bike activity from minute 550 to 561"*. Additionally, user-logged mood data with corresponding timestamps is integrated, such as *"The person logged their mood as*

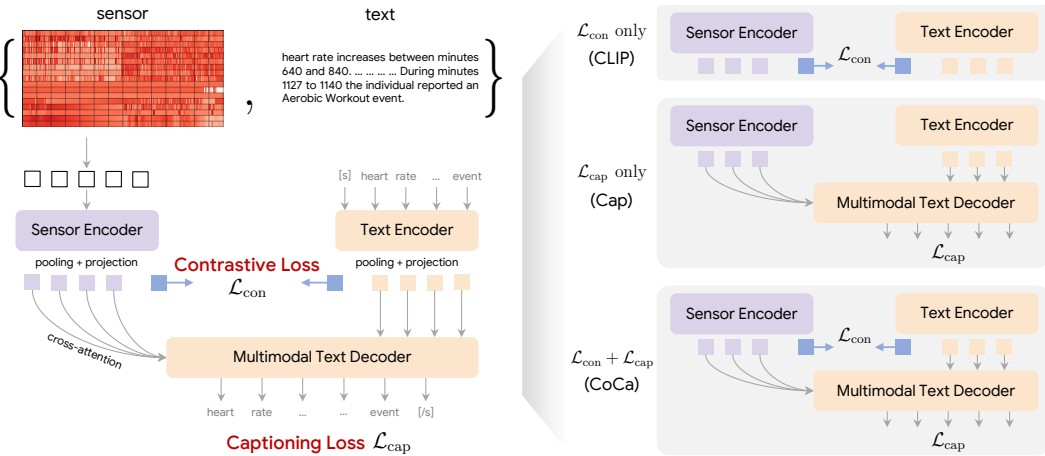

Figure 3: **The `SensorLM` architecture, pretraining objectives, and representative variants.** Our framework integrates both contrastive ($\mathcal{L}_{\text{con}}$) and generative ($\mathcal{L}_{\text{cap}}$) objectives, extending prior approaches (e.g., CLIP [25], Cap [33], CoCa [39]) to sensor data and recovering them as specific configurations within a single architecture.

*Frustrated at minute 1110."* Integrating this contextual information enables the model to associate real-world events and subjective states with underlying sensor patterns.

### 3.3 Large-Scale Pretraining Sensor-Text Dataset

Building upon the collected sensor data and caption generation pipeline, we construct the largest sensor-language paired pretraining dataset to date – **orders of magnitude** larger than prior studies (Table 1). All data are de-identified and used with participant consent and IRB review (`Advarra`). The dataset comprises 2,489,570 person-days of data from 103,643 people across 127 countries, collected between March 1$^{\text{st}}$ and May 1$^{\text{st}}$, 2024. (Further details are provided in Appendix A.1.)

We generate multiple caption combinations and evaluate their effectiveness in Sec. 5. Unless otherwise specified, we use the combination of structural and semantic captions in our main experiments.

## 4 SensorLM

`SensorLM` presents a generic framework for training sensor-language foundation models (Fig. 3). Inspired by prominent VLP architectures with contrastive (e.g., CLIP [25], SigLIP [42]), generative (e.g., Cap [33]), and hybrid (e.g., CoCa [39]) objectives, `SensorLM` extends these approaches to sensor data and recovers them as specific configurations within a single architecture.

**From Unimodal to Multimodal Paradigms.** We first position `SensorLM` within the broader family of foundation models that leverage language supervision, adapting these paradigms to the sensor domain. Traditional discriminative single-encoder models perform well on fixed-vocabulary tasks like activity recognition but lack flexibility for open-ended language grounding. Contrastive dual-encoder approaches [25] enable zero-shot generalization and efficient retrieval by aligning modalities in a shared space. Encoder-decoder architectures, in contrast, generate rich natural language conditioned on the input (e.g., sensor data), offering fine-grained interpretability via language. While expressive, they may struggle with robust cross-modal alignment. `SensorLM` integrates these complementary strategies into a single framework for sensor-language pretraining.

**Architecture.** `SensorLM` comprises a sensor encoder, a text encoder, and a multimodal text decoder (Fig. 3). The *sensor encoder* transforms time-series sensor data into a compact latent representations. We adapt a Vision Transformer (ViT) architecture to handle sensor data by segmenting the sequence into patches, applying linear embeddings, and processing them through transformer blocks to capture both local and long-range temporal dependencies. The *text encoder* follows a similar process to encode input text into unimodal representations. Finally, the *multimodal text decoder* is a causally masked transformer that integrates both sensor features (via cross-attention) and text features to produce multimodal text representations.

**Pretraining Objectives.** `SensorLM` is pretrained with a composite loss function that combines both *contrastive* and *generative* objectives. Given a batch of $N$ sensor-text pairs $\{(\boldsymbol{x}_n, \boldsymbol{y}_n)\}_{n \in [N]}$, the *contrastive loss* $\mathcal{L}_{\text{con}}$ is applied to normalized unimodal text embeddings $\boldsymbol{v}_i$ (from the text encoder) and sensor embeddings $\boldsymbol{s}_i$ (from the sensor encoder) via a symmetric cross-modal objective [5]:

$$\mathcal{L}_{\text{con}} = -\frac{1}{N}\left( \underbrace{\sum_{i=1}^{N} \log \frac{\exp(\text{sim}(\boldsymbol{s}_i, \boldsymbol{v}_i)/\tau)}{\sum_{j=1,\ j\neq i}^{N} \exp(\text{sim}(\boldsymbol{s}_i, \boldsymbol{v}_j)/\tau)}}_{\text{sensor-to-text}} + \underbrace{\sum_{i=1}^{N} \log \frac{\exp(\text{sim}(\boldsymbol{v}_i, \boldsymbol{s}_i)/\tau)}{\sum_{j=1,\ j\neq i}^{N} \exp(\text{sim}(\boldsymbol{v}_i, \boldsymbol{s}_j)/\tau)}}_{\text{text-to-sensor}} \right),$$

where $\text{sim}(\cdot, \cdot)$ is the similarity measure between embeddings, and $\tau$ denotes the temperature parameter. The *captioning loss* $\mathcal{L}_{\text{cap}}$ is a standard cross-entropy loss applied to the outputs of multimodal text decoder. It maximizes the conditional likelihood of the paired text $\boldsymbol{y}$ given the sensor input $\boldsymbol{x}$ under a forward autoregressive factorization:

$$\mathcal{L}_{\text{cap}} = -\sum_{t=1}^{T} \log \mathbb{P}_\theta(\boldsymbol{y}_t \mid \boldsymbol{y}_{<t}, \boldsymbol{x}).$$

**Combined Objective and Representative Variants.** The final objective is a weighted combination: $\mathcal{L}_{\texttt{SensorLM}} = \lambda_{\text{con}} \cdot \mathcal{L}_{\text{con}} + \lambda_{\text{cap}} \cdot \mathcal{L}_{\text{cap}}$, where $\lambda_{\text{con}}$ and $\lambda_{\text{cap}}$ control the balance between contrastive and generative learning. Under this unified formulation, `SensorLM` seamlessly recovers representative variants within a single framework (Fig. 3): (1) CLIP (i.e., $\lambda_{\text{cap}} = 0$), (2) Cap (i.e., $\lambda_{\text{con}} = 0$), and (3) CoCa (i.e., $\lambda_{\text{con}} = \lambda_{\text{cap}} = 1$). We investigate these model variants in Sec. 5.3.

**The `SensorLM` Family.** We train four variants of `SensorLM` with increasing sizes: `SensorLM-S`, `SensorLM-B`, `SensorLM-L`, and `SensorLM-XL`. These variants employ sensor encoders with 3M, 114M, 404M, and 1.27B parameters (details in Appendix B.1). We study the effects of model scaling in Sec. 5. Unless otherwise specified, we report results using `SensorLM-B`.

## 5 Experiments & Results

**Datasets.** We utilize three evaluation datasets for activity recognition, health-related tasks and cross-modal retrieval. Detailed information and demographic breakdowns are provided in Appendix A.3.

- *Activity dataset.* The Activity dataset comprises 22,289 person-days from 10,013 individuals. We randomly sampled ∼1,000 test examples for each activity for zero-shot activity recognition (AR) and few-shot adaptation. This dataset is from the same population as our pretraining sensor data.
- *Metabolic dataset.* The metabolic dataset, used for "*Hypertension*" and "*Anxiety*" prediction tasks, were sourced from a held-out, IRB-approved observational study of adults in the US [22]. The data comprises 241,532 examples (151,346 training and 90,186 testing) from 1,979 individuals. It includes self-reported information covering demographics (e.g., age, sex, weight) and medical conditions (e.g., "*Hypertension*" and "*Anxiety*"), providing rich labels for health analyses.
- *Sensor-text retrieval dataset.* We construct a retrieval dataset of 39,766 examples sourced from a separate set of 975 individuals who were not included in the pretraining set. The same selection criteria and caption generation method used in pretraining were applied.

**Baselines.** For zero-shot classification and cross-modal retrieval, there are currently no existing multimodal baselines with these capabilities. We therefore compare against two LLMs, Gemini 2.0 Flash [29] and Gemma-3-27B [30], by formatting sensor data as tabular input. In addition, we finetune Gemini 2.0 with paired sensor and activity data through supervised fine-tuning (SFT). For few-shot learning and linear probing, we compare with SOTA SSL methods including SimCLR [5], DINO [3], and Masked Siamese Network (MSN) [2]. Implementation details can be found in Appendix B.

**Metrics.** For zero-shot classification, we report area under the ROC curve (AUROC), Macro F1-score, and Balanced Accuracy (BAcc). For few-shot learning, we report AUROC across 5, 10, 20, 50 samples per class. Cross-modal retrieval is evaluated using Recall@1 and Recall@5 (R@K).

### 5.1 Main Results

**Zero-Shot Learning.** One of the key capabilities enabled by `SensorLM` is zero-shot classification, which we evaluate on a range of AR tasks. We use prompt ensembling [25] from the text encoder

Table 2: **Zero-shot activity and concept classification.** We compare `SensorLM` with representative LLM baselines across **(a)** main activity classification, **(b-d)** activity-related concept classification, and **(e-f)** fine-grained recognition. Gemini 2.0 with supervised fine-tuning (SFT) does not output class-probability scores; therefore AUROC is not reported ("−"). Detailed task definitions and experimental setups are provided in Appendix B.5.

#### (a) Activity recognition (20-class)

| Metrics | AUROC$^\uparrow$ | F1$^\uparrow$ | BAcc$^\uparrow$ |
|---|---|---|---|
| Gemma-3-27B [30] | 0.50 | 0.01 | 0.05 |
| Gemini 2.0 [29] | 0.51 | 0.03 | 0.07 |
| Gemini 2.0 (SFT) [29] | – | 0.06 | 0.10 |
| SensorLM | **0.84** (+33%) | **0.29** (+23%) | **0.31** (+21%) |

#### (b) Activity by environmental context

| Metrics | AUROC$^\uparrow$ | F1$^\uparrow$ | BAcc$^\uparrow$ |
|---|---|---|---|
| Gemma-3-27B [30] | 0.51 | 0.22 | 0.25 |
| Gemini 2.0 [29] | 0.50 | 0.19 | 0.25 |
| Gemini 2.0 (SFT) [29] | – | 0.25 | 0.28 |
| SensorLM | **0.64** (+13%) | **0.33** (+08%) | **0.38** (+10%) |

#### (c) Cardio *vs.* strength training

| Metrics | AUROC$^\uparrow$ | F1$^\uparrow$ | BAcc$^\uparrow$ |
|---|---|---|---|
| Gemma-3-27B [30] | 0.53 | 0.42 | 0.49 |
| Gemini 2.0 [29] | 0.50 | 0.39 | 0.50 |
| Gemini 2.0 (SFT) [29] | – | 0.44 | 0.53 |
| SensorLM | **0.71** (+18%) | **0.63** (+19%) | **0.66** (+13%) |

#### (d) Locomotion *vs.* stationary

| Metrics | AUROC$^\uparrow$ | F1$^\uparrow$ | BAcc$^\uparrow$ |
|---|---|---|---|
| Gemma-3-27B [30] | 0.51 | 0.40 | 0.51 |
| Gemini 2.0 [29] | 0.55 | 0.52 | 0.55 |
| Gemini 2.0 (SFT) [29] | – | 0.53 | 0.55 |
| SensorLM | **0.61** (+06%) | **0.58** (+05%) | **0.58** (+03%) |

#### (e) Fine-grained recognition (gym cardio)

| Metrics | AUROC$^\uparrow$ | F1$^\uparrow$ | BAcc$^\uparrow$ |
|---|---|---|---|
| Gemma-3-27B [30] | 0.49 | 0.16 | 0.25 |
| Gemini 2.0 [29] | 0.52 | 0.20 | 0.26 |
| Gemini 2.0 (SFT) [29] | – | 0.28 | 0.35 |
| SensorLM | **0.76** (+24%) | **0.50** (+22%) | **0.51** (+16%) |

#### (f) Fine-grained recognition (outdoor sports)

| Metrics | AUROC$^\uparrow$ | F1$^\uparrow$ | BAcc$^\uparrow$ |
|---|---|---|---|
| Gemma-3-27B [30] | 0.50 | 0.18 | 0.25 |
| Gemini 2.0 [29] | 0.54 | 0.22 | 0.29 |
| Gemini 2.0 (SFT) [29] | – | 0.26 | 0.32 |
| SensorLM | **0.83** (+29%) | **0.52** (+26%) | **0.53** (+21%) |

with diverse prompts to compute average label embeddings, and classify by selecting the label whose embedding is closest to the sensor embedding. We include 6 tasks in total, including *20-class activity recognition*, *activity concept classification* (2-4 classes by "environmental" context or "physiological" type), and *fine-grained recognition* (4 classes within "gym cardio" or "outdoor sports"). As shown in Table 2, LLM baselines perform near random on sensor-based activity classification. In contrast, `SensorLM` demonstrates strong zero-shot capabilities across all metrics and tasks. For abstract concepts such as "environmental" or "physiological" type, the superior performance suggests that the model captures both explicit activity categories and *higher-level conceptual* understanding.

**Zero-Shot Cross-Modal Retrieval.** We evaluate the zero-shot cross-modal retrieval performance of `SensorLM` by assessing its ability to retrieve relevant text descriptions given sensor queries (Sensor → Text) and vice versa (Text → Sensor). Sensor-to-text retrieval enables querying description based on sensor input, while text-to-sensor retrieval supports use cases such as expert-driven querying for specific sensor patterns using natural language. Table 3 confirms `SensorLM`'s exceptionally strong performance across all sample sizes, significantly outperforming LLM baselines, which largely failed at this task. `SensorLM` achieves perfect retrieval on a 100-sample benchmark and maintains high accuracy even at larger scales (5k and 40k samples; full results in Appendix C.2).

Qualitatively, Fig. 4 verifies `SensorLM`'s ability to retrieve accurate descriptions for unseen sensor data involving multiple activities. In addition to correctly retrieving the ground truth, the top similar captions are also semantically relevant – capturing either similar activities occurring at the same time (e.g., "Walk" *vs.* "Run") or the same activity at different times. This reflects its understanding of both activity semantics and temporal alignment.

Table 3: **Zero-shot cross-modal retrieval.** `SensorLM` achieves consistently strong retrieval performance on a held-out dataset. Baseline LLMs struggle, with most tasks infeasible due to context limits (marked as "−"). Detailed experimental setups and complete results are in Appendix B.7 and C.2.

| Metrics | 100 samples | | 5,000 samples | |
|---|---|---|---|---|
| | R@1 | R@5 | R@1 | R@5 |
| *Sensor → Text:* | | | | |
| Gemma-3-27B [30] | 1.0 | 5.0 | – | – |
| Gemini 2.0 [29] | 5.0 | 9.0 | – | – |
| SensorLM | **100.0** | **100.0** | **98.2** | **99.4** |
| *Text → Sensor:* | | | | |
| Gemma-3-27B [30] | – | – | – | – |
| Gemini 2.0 [29] | – | – | – | – |
| SensorLM | **100.0** | **100.0** | **96.7** | **99.2** |

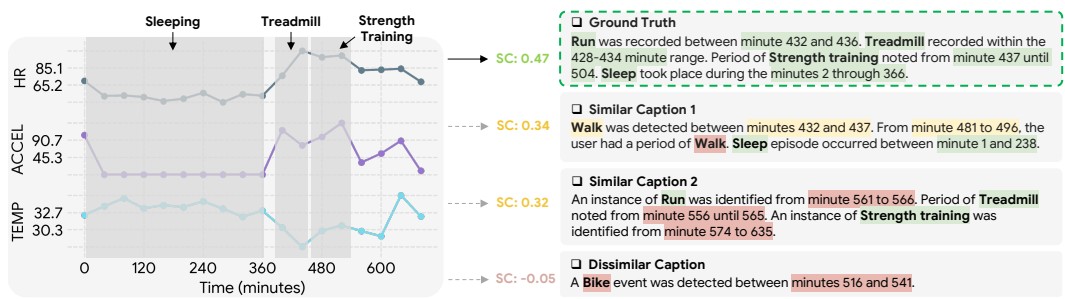

Figure 4: **Zero-shot sensor-to-text retrieval qualitative example.** We show similarity scores (SC) for the correctly retrieved ground truth, top similar captions, and a dissimilar caption. Green highlights correct events and time frames; Yellow indicates partial matches; and Red denotes incorrect events and time frames. In addition to correctly retrieving the ground truth, `SensorLM` also demonstrates semantic understanding by assigning high similarity scores to partially correct candidates.

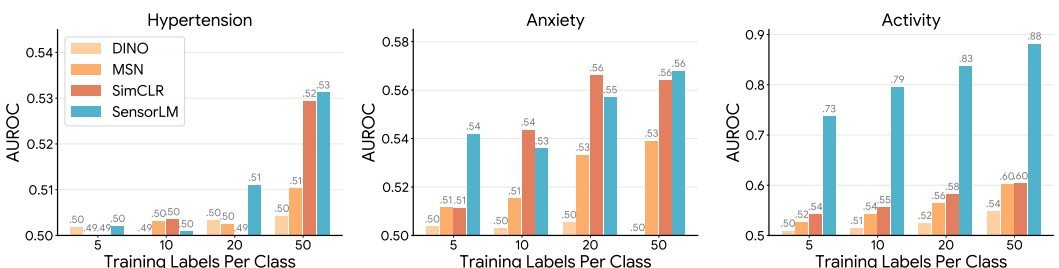

Figure 5: **Few-shot adaptation to downstream tasks.** We evaluate the label efficiency of different pretrained sensor encoders with varying numbers of training labels per class. Across diverse tasks, `SensorLM` achieves comparable or superior performance than SOTA methods. Additional results are provided in Appendix C.1.

**Few-Shot Transfer Learning.** We evaluate the quality of the learned sensor representations through few-shot learning on three tasks: 20-class AR using the *Activity* dataset, and two binary prediction tasks on clinical condition ("Hypertension") and mental health ("Anxiety") using the *Metabolic* dataset. Fig. 5 shows the few-shot performance of `SensorLM` compared to baseline SSL methods as the number of training labels per class increases from 5 to 50. For AR, `SensorLM` significantly out-performs baselines across all few-shot settings, achieving an AUROC of $0.88$ with only 50 labels per class, highlighting its effective representations in discriminating activities under limited supervision. For "Hypertension" and "Anxiety" prediction, `SensorLM` shows competitive performance. Linear probing results on the full training set support similar conclusions (Appendix C.1).

## 5.2 Analyses

**Scaling Laws.** We investigate the scaling behavior of `SensorLM` when trained on large-scale sensor-language dataset by varying *training compute*, *dataset size*, and *model scale* [12]. As observed in Fig. 6, the performance on zero-shot activity recognition improves consistently with increased training steps and data size. For example, when scaling up the compute, `SensorLM-B` increases from an AUROC of 0.66 at 17.4 TPU hours to 0.75 at 174 TPU hours. Expanding training data also yields consistent gains, especially for larger models. However, gains beyond 12 million hours diminish, showing saturation trends as reported in prior work [12, 24]. Larger models such as `SensorLM-L` and `SensorLM-XL` achieve higher performance when sufficiently trained, though under low compute they may underperform due to undertraining (a phenomenon consistent with scaling law observations). Overall, `SensorLM` demonstrates consistent and predictable scaling behaviors, highlighting the applicability of scaling laws to sensor-language modeling.

**Robustness to Out-of-Domain and Incomplete Data (Appendix C.7).** We evaluate `SensorLM`'s robustness to out-of-domain (OOD) and missing data. With our metabolic downstream dataset containing additional device types unseen during pretraining, we compare performance on in-domain and OOD devices. `SensorLM` demonstrates strong generalization with only a degradation in some of

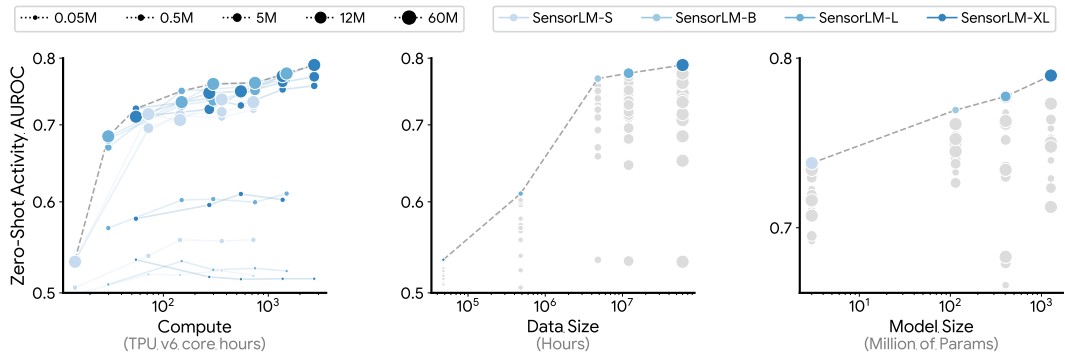

Figure 6: **Scaling behaviors of** `SensorLM`. We show the zero-shot downstream performance of `SensorLM` as a function of compute (**left**), data size (**middle**), and model size (**right**). Results demonstrate that increasing compute, data, and model size each leads to consistent performance gains.

the tasks. We also assess missing-data scenarios, finding that performance degrades only marginally, indicating that `SensorLM` remains robust and flexible with incomplete sensor inputs.

**Zero-Shot Generalization to Unseen Classes.** To evaluate `SensorLM`'s generalizability to novel activities, we conducted a case study by pre-training on data comprising 20 activity classes and tested on previously unseen ones (details in Appendix B.8). The results of one-versus-all AUROC and 95% CIs for these classes are shown in Fig. 7(a). The impressive zero-shot performance achieved on these unseen activities indicates that `SensorLM` learns the broader concept of activities rather than memorizing the training set. Specifically, activities that are conceptually similar to those in the training set exhibit higher performance (e.g., "Snowboarding" is similar to "Skiing"). This is further supported by the visualization of label and sensor embeddings using t-SNE [34] (Fig. 7(b)), where semantically related activities form coherent clusters, indicating that `SensorLM` infers unseen classes based on their proximity to known concepts in the learned embedding space.

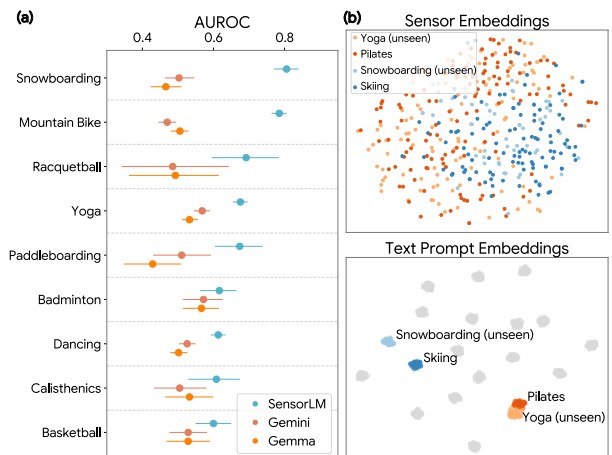

Figure 7: **Zero-shot generalization analysis of** `SensorLM`. **(a)** Overall performance on unseen activities. **(b)** Learned representations in both the sensor and text embedding spaces. We visualize conceptually similar but previously unseen activities as a case study. The semantic alignment across modalities allows `SensorLM` to infer the nature of unseen activities based on their conceptual proximity to known ones.

**Sensor Caption Generation.** A key strength of `SensorLM` stems from its encoder-decoder architecture, trained with a generative objective. This design positions `SensorLM` to effectively process multimodal sensor embeddings, yielding *sensor captioning* capabilities. Beyond its inherent classification and retrieval capabilities, `SensorLM` demonstrates strong generative performance. On a dedicated evaluation set of 200 sensor-text pairs (sampled from the sensor-caption retrieval dataset), `SensorLM` directly generates captions from sensor input (and a `[START]` token). As shown in Fig. 8, `SensorLM` (pre-trained on semantic captions) outperforms powerful LLM baselines, including Gemini 2.0 Flash [29] and Gemma-3-27B [30]. These results highlight `SensorLM`'s capability as a sensor-language foundation model that can both interpret sensor data and generate meaningful natural language descriptions.

## 5.3 Ablation Studies

We provide main ablation studies on `SensorLM`, with complete results in Appendix C.4 and C.5.

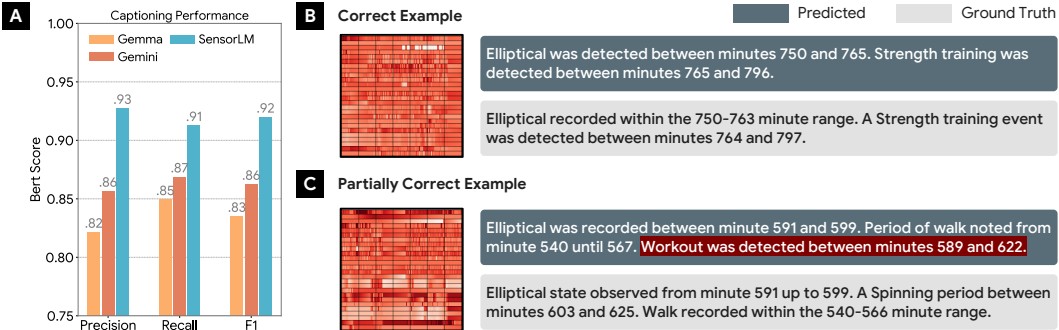

Figure 8: **Sensor caption generation results. (A)** Captioning performance of `SensorLM` and baselines. For `SensorLM`, only the sensor data and a `[START]` token are provided to generate captions. **(B)** A correctly generated example. **(C)** A partially correct example with inaccurate components. `SensorLM` produces meaningful and semantically accurate captions. Detailed experimental setups and additional results are in Appendix B.9 and C.3.

**Understanding the impact of pretraining caption variants.** We investigate how different levels of sensor descriptions affect downstream tasks. Table 4 shows that semantic captions are crucial for zero-shot AR, whereas models trained solely on statistical or structural captions underperform due to a lack of activity-level concepts. Combining semantic and structural captions improves zero-shot AR over semantic-only models, indicating that local signal patterns and temporal structure contribute complementary information. However, adding statistical captions to semantic ones reduces zero-shot AR performance, likely because daily-level statistics are less relevant to fine-grained activities. Yet, statistical captions benefit "Anxiety" and "Hypertension" predictions, either alone or when combined with other caption types. Overall, the "semantic + structural" combination provides the best trade-off across tasks.

**Pretraining architectural variants.** We study different architectural variants trained with contrastive and generative objectives. As Table 5 reports, `SensorLM` (CoCa) consistently outperforms the single-objective variants across key metrics for both zero-shot classification and linear probing. The generative objective enhances the contrastive alignment by providing fine-grained text supervision, leading to improved cross-modal understanding. Notably, the `SensorLM` (Cap) model performs poorly in zero-shot classification due to the absence of a trained projection head.

Table 4: **Comparisons of `SensorLM` caption variants.** We evaluate different combinations of caption types used during pretraining. AUROC$^{\uparrow}$ is used as the metric. Best results of each column are in **bold** and the second best are underlined. Default setting used in the main experiments is marked in gray.

| Caption Variant | | | Zero-Shot | Linear Probing | |
|---|---|---|---|---|---|
| statistical | structural | semantic | Activity | Activity | Anxiety |
| ✓ | ✗ | ✗ | 0.51 | 0.76 | 0.67 |
| ✗ | ✓ | ✗ | 0.50 | 0.78 | 0.63 |
| ✗ | ✗ | ✓ | 0.71 | **0.95** | 0.65 |
| ✓ | ✗ | ✓ | 0.66 | 0.84 | **0.68** |
| ✗ | ✓ | ✓ | **0.84** | 0.94 | 0.65 |
| ✓ | ✓ | ✗ | 0.49 | 0.79 | 0.67 |
| ✓ | ✓ | ✓ | 0.66 | 0.86 | **0.68** |

Table 5: **Comparisons of `SensorLM` architectural variants.** We compare different choices used during pretraining. Default setting used in the main experiments is marked in gray.

| Arch Variant | Zero-Shot | | Linear Probing | |
|---|---|---|---|---|
| | AUROC$^{\uparrow}$ | F1$^{\uparrow}$ | AUROC$^{\uparrow}$ | F1$^{\uparrow}$ |
| `SensorLM` (CLIP) | 0.83 | **0.29** | 0.93 | 0.53 |
| `SensorLM` (SigLIP) | 0.78 | 0.17 | 0.87 | 0.38 |
| `SensorLM` (Cap) | 0.55 | 0.01 | 0.90 | 0.32 |
| `SensorLM` (CoCa) | **0.84** | **0.29** | **0.94** | **0.57** |

## 6  Discussion

**Limitations.** It is crucial to understand that `SensorLM` is not a clinically validated diagnostic tool and is not intended for clinical diagnosis, treatment, or medical decision-making; deployment for such uses would require further analysis of applicable healthcare regulations. Additionally, our evaluation is limited to specific wearable devices and sensor data; while the method is general, further work is needed to explore its generalizability on other types of sensor data.

**Conclusion.** We present `SensorLM`, a family of sensor-language foundation models that unlock the understanding of wearable sensor data through natural language, enabled by a novel hierarchical captioning pipeline and the largest sensor-language dataset to date. `SensorLM` achieves superior performance in zero-shot, few-shot, and cross-modal retrieval tasks, as well as demonstrating intriguing properties such as scaling, generative, and zero-shot generalization capabilities.

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

Table 6: **Definitions of sensor features.** We detail the names, units, and definitions of the 26 features derived from PPG, accelerometer, skin conductance, skin temperature, and altitude signals.

| Feature | Unit | Definition |
|---|---|---|
| ***Photoplethysmography*** (PPG) | | |
| Heart Rate | Beats/Min | Mean of instantaneous heart rate. |
| Shannon Ent. RR | Nats | Shannon entropy of the RR intervals. |
| Shannon Ent. RR Diffs | Nats | Shannon entropy of the RR interval differences. |
| RMSSD | Msec | Root mean squared st. dev. of RR intervals. |
| SDNN | Msec | Standard deviation of RR intervals. |
| RR Percent Valid | % | % of 5-minute window with valid RR intervals. |
| RR $80^{th}$ Percentile | Msec | $80^{th}$ percentile of 5-minute window of RR intervals. |
| RR $20^{th}$ Percentile | Msec | $20^{th}$ percentile of RR intervals. |
| RR Median | Msec | Median RR interval. |
| Heart Rate at Rest | Beats/Min | Mean of heart rate at rest. |
| ***Accelerometer*** (ACC) | | |
| Step Count | Steps | Number of steps. |
| Jerk Autocorrelation Ratio | a.u. | Ratio of lag=1 autocorrelation to energy in 1st 3-axis principal component. |
| Log Energy | a.u. | Log of sum of 3-axis root mean squared magnitude. |
| Covariance Condition | a.u. | Estimate of condition number for 3-axis covariance matrix. |
| Log Energy Ratio | a.u. | Log of ratio of sum of energy in 1st 3-axis principal component over energy of 3-axis root mean squared magnitude. |
| Zero Crossing St.Dev. | Seconds | Standard deviation of time between zero crossing of 1st 3-axis principal component. |
| Zero Crossing Average | Seconds | Mean of time between zero crossing of 1st 3-axis principal component. |
| Axis Mean | a.u. | Mean of 3-axis. |
| Kurtosis | a.u. | Kurtosis of 3-axis root mean squared magnitude. |
| Sleep Coefficient | a.u. | Sum of 3-axis max-min range, binned into 16 log-scaled bins. |
| ***Skin Conductance*** (EDA) | | |
| Skin Conductance Value | $\mu$Siemens | Center of linear tonic SCL value fit. |
| Skin Conductance Slope | $\mu$S/Min | Intraminute slope of SCL values. |
| Lead Contact Counts | Counts | Number of times leads of the sensor contacting wrist in a minute. |
| ***Skin Temperature*** (TEMP) | | |
| Skin Temperature Value | °C | Value of skin temperature. |
| Skin Temperature Slope | °C/Min | Slope of skin temperature. |
| ***Altimeter*** (ALT) | | |
| Altitude St.Dev. Norm | Hectopascals | Standard deviation of altimeter readings. |

# A  Dataset Overview

## A.1  Pretraining Sensor Dataset

Our wearable devices incorporate five distinct sensors: *Photoplethysmography* (PPG), *Accelerometer* (ACC), *Skin Conductance* (electrodermal activity, EDA), *Skin Temperature* (TEMP), and *Altitude* (ALT). While these sensors collect raw waveform signals at high frequencies (i.e., 100Hz, 25Hz, 200Hz, 6Hz, and 10Hz, respectively), we do not directly utilize these high-resolution signals. This decision is driven by practical constraints, including prohibitive storage requirements and battery consumption, which make raw data storage infeasible at our scale. Moreover, learning from full-day raw waveforms is computationally impractical; for example, a 200Hz signal over 24 hours results in approximately 17 million time points per instance.

Instead, we curate minutely aggregated features from the raw signals and use them as model inputs. These features are grounded in established domain literature, with prior work demonstrating their clinical utility [24]. For example, heart rate variability metrics like *RMSSD* and the *Shannon Entropy of RR Intervals* are recognized markers of cardiovascular health, while accelerometer-derived features

Table 7: **Detailed statistics and demographic information for the pretraining and downstream datasets.** We report participant counts for the pretraining, *Activity* (downstream), and *Metabolic* (downstream) datasets, along with demographic breakdowns by sex, age, and BMI.

| Category | Pretraining | Activity (downstream) | | Metabolic (downstream) | |
|---|---|---|---|---|---|
| | | train | test | train | test |
| *Sex:* | | | | | |
| Male | 64,194 | 27,653 | 6,092 | 551 | 258 |
| Female | 39,376 | 10,145 | 2,248 | 670 | 455 |
| Not Specified | 73 | 24 | 3 | 0 | 0 |
| *Age:* | | | | | |
| $18 - 39$ | 52,004 | 19,340 | 4,492 | 415 | 223 |
| $40 - 59$ | 41,296 | 15,309 | 3,172 | 637 | 384 |
| $60 - 79$ | 9,340 | 2,875 | 618 | 198 | 121 |
| $\geq 80$ | 676 | 120 | 31 | 0 | 1 |
| Not Specified | 327 | 30 | 0 | 0 | 0 |
| *BMI:* | | | | | |
| Healthy ($< 25$) | 38,582 | 15,942 | 3,685 | 319 | 188 |
| Overweight ($25 - 30$) | 34,188 | 14,154 | 3,017 | 343 | 206 |
| Obese ($\geq 30$) | 25,969 | 6,131 | 1,316 | 481 | 274 |
| Not Specified | 324 | 81 | 18 | 49 | 28 |
| **Total** | 103,643 | 37,822 | 8,343 | 1,250 | 729 |

such as *Jerk Ratio* capture movement quality. Descriptions of the derived features and their associated sensor modalities are provided in Table 6. Our pretraining set included **18** smartwatch and fitness-tracker models (*Google Pixel Watch 1-3*, *Fitbit MobileTrack*, *Charge 4-6*, *Sense 1-2*, *Versa 2-4*, *Inspire 2-3*, *Luxe*, *Versa*, *Charge 2*, and *Alta HR*).

Table 7 summarizes the demographic composition of our pretraining dataset. The only inclusion criterion was having valid sensor data for at least 20% of one day and at least one logged event. The cohort is 38% female with a mean age of 41.7 years (range: $18 - 100$). BMI distributions indicate 37% healthy ($< 25$), 33% overweight ($25 - 30$), and 25% obese ($\geq 30$), with the remaining 5% unspecified.

All data are de-identified and used with participant consent and IRB review (`Advarra`). The use of data for pretraining in this manner was approved as exempt under 45 CFR § 46.104(d)(4) *"because the research involves the use of identifiable private information/biospecimens; and information, which may include information about biospecimens, is recorded by the investigator in such a manner that the identity of the human subjects cannot readily be ascertained directly or through identifiers linked to the subjects, the investigator does not contact the subjects, and the investigator will not re-identify subjects."*

## A.2 Sensor Caption Generation

To encourage linguistic diversity and prevent overfitting to a single text pattern, we employ a variety of sentence templates across the three levels of sensor captions. This section details the design of these templates.

Table 8 provides a selection of **statistical caption** templates, illustrated using "Heart rate" as the example feature with an average of 88.7, standard deviation of 9.3, minimum of 70.8, and maximum of 134.9. We employ 20 rewrite templates in total to generate varied statistical descriptions.

Table 9 shows examples of **structural caption** templates, including descriptions of trends (e.g., "*a decreasing trend in heart rate between minute 680 and 960*") and spike events (e.g., "*a spike in step count at minute 720*"). We use 15 rewrite templates in total for structural captions.

Table 8: **Example of prompt templates used in statistical captions.**

```
Statistical Caption Templates

1. The average Heart rate value is 88.7, with extremes at 134.9 (max)
   and 70.8 (min), and a std of 9.3.
2. The Heart rate data exhibits a mean of 88.7, a standard deviation of
   9.3, and its extreme values are 70.8 and 134.9.
3. Heart rate average 88.7, reaching a maximum of 134.9 and a minimum of
   70.8, with a standard deviation of 9.3.
4. Heart rate exhibits a mean of 88.7, with peak and minimal values
   reaching 134.9 and 70.8, and a standard deviation of 9.3.
5. For the Heart rate measurements, the mean is 88.7, the standard
   deviation is 9.3, and the data lies between 70.8 and 134.9.
......
```

Table 9: **Example of prompt templates used in structural captions.**

```
Structural Caption Templates

Trends:

1. An decreasing trend in Heart rate data recorded between minute 680
   and 960.
2. Heart rate exhibits decreasing trend during minute 680-960 interval.
3. An decreasing trend in Heart rate data recorded between minute 680
   and 960.
4. The Heart rate trend from minute 680 to 960 is decreasing.
5. From minute 680 to 960, Heart rate exhibits an decreasing trend.
......

Spikes:

1. Spike event recorded for steps at minute 720.
2. Data indicates a peak for steps at the 720-minute mark.
3. Minute 720 shows a spike for the steps.
4. A peak is detected for steps at minute 720.
5. The steps experienced a spike at minute 720.
......
```

Table 10 presents templates for **semantic captions**, which describe high-level activities and their associated timeframes, such as "*Outdoor Bike recorded between minute 1121 and 1133*". We use 20 rewrite templates in total for semantic captions.

## A.3 Downstream Datasets

Table 7 summarizes the demographic distributions for the downstream linear probing and few-shot evaluation datasets: the *Activity dataset* and the *Metabolic dataset*. For zero-shot classification tasks, we resample the test data to create a more balanced evaluation set. Additional details, including the number of samples per class, are provided in Appendix B.5.

The activity downstream dataset used these same devices as our pretraining set, while the metabolic downstream dataset incorporated an additional **17** device types (*Fitbit Flex*, *One*, *Zip*, *Charge*, *Charge HR*, *Charge 3*, *Flex 2*, *Inspire HR*, *Blaze*, *Alta*, *Fitbit Ultra*, *Versa Lite*, *Surge*, *Ionic*, *Fitbit Classic*, *Ace 2*, *Inspire*).

Table 10: **Example of prompt templates used in semantic captions.**

```
Semantic Caption Templates

  1. From minute 1121 to 1133, the user had a period of Outdoor Bike.
  2. Outdoor Bike recorded within the 1121-1133 minute range.
  3. Outdoor Bike episode occurred between minute 1121 and 1133.
  4. Outdoor Bike was recorded between minute 1121 and 1133.
  5. Identified Outdoor Bike across the timeframe of minute 1121 to 1133.
  ......
```

Table 11: **The** `SensorLM` **family**. `SensorLM` consists of four variants with increasing model sizes. Architectural details for the sensor encoder, text encoder, and multimodal text decoder are provided.

| Model | Sensor Encoder | | | Text Encoder & Multimodal Decoder | | | | Sensor / Text | |
|---|---|---|---|---|---|---|---|---|---|
| | layers | MLP | # params | layers$_{enc}$ | layers$_{dec}$ | MLP | # params | hidden | heads |
| SensorLM-S | 12 | 512 | 3M | 12 | 3 | 512 | 12M | 128 | 16 |
| SensorLM-B | 12 | 3072 | 114M | 12 | 3 | 3072 | 191M | 768 | 12 |
| SensorLM-L | 24 | 4096 | 404M | 24 | 3 | 4096 | 519M | 1024 | 16 |
| SensorLM-XL | 40 | 5632 | 1.27B | 40 | 3 | 5632 | 1.45B | 1408 | 16 |

# B   Implementation Details

## B.1   SensorLM Model Architecture

As described earlier, `SensorLM` consists of a sensor encoder, a text encoder, and a multimodal text decoder (Fig. 3). The input to the sensor encoder is a matrix of shape [26 features × 1440 minutes], representing one-day windows of sensor data at minutely resolution. The sensor encoder is built using a ViT-2D backbone with a 2D patch size of $(2, 10)$, producing 1872 tokens.

The text encoder and multimodal text decoder follow standard transformer architectures. The multimodal text decoder attends to the output tokens of the sensor encoder via cross-attention [39], enabling an encoder-decoder setup for caption generation. For contrastive learning, we use representations from the sensor encoder and unimodal text encoder. Both representations are average-pooled, passed through projection heads, normalized, and then used to compute the contrastive loss $\mathcal{L}_{con}$ between sensor embeddings $s_i$ and text embeddings $v_i$.

The `SensorLM` family consists of four variants with different (increasing) model sizes: `SensorLM-S`, `SensorLM-B`, `SensorLM-L`, and `SensorLM-XL`. Architectural specifications for each variant are provided in Table 11.

## B.2   Pretraining Details

All models are trained using the `SensorLM` pretraining objective ($\mathcal{L}_{\texttt{SensorLM}}$) on Google v6 TPUs for 50k steps. We use a batch size of 1024 sensor-text pairs, with $\lambda_{con} = \lambda_{cap} = 1$ for main experiments. The temperature $\tau$ for the contrastive loss is set to 0.01. Optimization is performed using Adam optimizer with $\beta_1 = 0.9$ and $\beta_2 = 0.95$. A cosine warm-up schedule is applied for the first 10% of training steps, followed by linear decay of the learning rate to zero.

## B.3   Self-Supervised Learning Baselines

We compare `SensorLM` against the following self-supervised learning (SSL) baselines:

- **SimCLR [5]** is a widely adopted contrastive learning framework that aligns representations from augmented views of the same input while repelling those of different inputs. It relies solely

Table 12: **Hyperparameters for self-supervised learning baselines.** We provide settings used for MSN [2], DINO [3], and SimCLR [5]. A single row value indicates that the setting was applied to all methods.

| Configuration | MSN [2] | DINO [3] | SimCLR [5] |
|---|---|---|---|
| training steps | | 50,000 | |
| warmup steps | | 2,500 | |
| optimizer | | AdamW [18] | |
| opt. momentum $[\beta_1, \beta_2]$ | | [0.9, 0.99] | |
| base learning rate | 0.001 | 0.004 | 0.001 |
| batch size | | 1,024 | |
| weight decay | | 0.0001 | |
| gradient clipping | | 3.0 | |
| learning rate schedule | | linear warmup & cosine decay | |
| data resolution | | 26 (sensor) $\times$ 1440 (minutes) | |

on contrastive loss and unlabeled data, and has proven effective for both visual and text-based representation learning.

- **MSN [2]** integrates contrastive-based pretraining with masked image modeling. MSN aligns the representation of a masked view with that of the unmasked input by processing only visible patches. This approach enhances scalability with Vision Transformers and yields semantically meaningful features, achieving strong performance in low-shot classification tasks.

- **DINO [3]** employs a self-distillation strategy using a teacher-student framework. The student network is trained to match the teacher's representations across different augmentations. DINO has demonstrated robust and generalizable feature learning across various domains.

All three baselines follow augmentation-driven training paradigms. We apply a standardized set of time-series augmentations (e.g., `jittering`, `scaling`, `time flipping`) based on prior work [17, 27, 28, 43]. Each augmentation is applied independently with a probability of 0.5. Jittering adds Gaussian noise with zero mean and a standard deviation uniformly sampled from $[0, 0.5]$. Scaling multiplies the input by a random factor sampled from $[1.1, 1.5]$. Notably, we omit scaling for DINO due to convergence issues observed during training.

All baseline models are pretrained from scratch under training settings listed in Table 12. For consistency, all methods use the same ViT-2D backbone as the sensor encoder in `SensorLM`, with a 2D patch size of $(2, 10)$.

## B.4 LLM Baselines

For zero-shot classification, cross-modal retrieval, and sensor captioning, we compare `SensorLM` against two LLMs: Gemini 2.0 Flash [29] and Gemma-3-27B [30], by formatting sensor data as tabular input.

- **Gemma-3-27B [30]** is a state-of-the-art open-weight large language model developed by Google DeepMind. Trained on a high-quality, multilingual corpus, it is optimized for instruction following and reasoning tasks. We use the instruction-tuned variant, which exhibits strong performance in both zero-shot and few-shot scenarios.

- **Gemini 2.0 Flash [29]** is a lightweight, high-performance member of Google's Gemini multimodal model family, designed for latency-sensitive applications. It supports vision-language reasoning and offers strong cross-modal capabilities with low inference latency.

For both baselines, we subsample the sensor data to fit within the context length limitations of these LLMs.

Table 13: **Definition of the activity-related concept classification tasks.**

| Class | Activities included | Prompt example |
|---|---|---|
| **Task: *Environmental context*** | | |
| Indoor Sports | Elliptical, Treadmill, Spinning, Weightlifting, Yoga, Core training, Pilates, Stairclimber, Dancing, Indoor climbing, Kickboxing | `User performed Indoor Sports.` |
| Outdoor Sports | Bike, Run, Hike, Tennis, Mountain Bike, Rollerblading, Golf | `User performed Outdoor Sports.` |
| Water Sports | Swim, Kayaking, Surfing, Paddleboarding | `User performed Water Sports.` |
| Winter Sports | Skiing, Snowboarding | `User performed Winter Sports.` |
| **Task: *Locomotion vs. stationary*** | | |
| Locomotion Exercise | Walk, Run, Hike, Stairclimber, Treadmill, Elliptical | `User performed Locomotion exercise.` |
| Stationary Exercise | Weightlifting, Yoga, Core training, Pilates | `User performed Stationary exercise.` |
| **Task: *Cardio vs. strength training*** | | |
| Cardio Exercise | Elliptical, Treadmill, Spinning, Stairclimber, Run | `User performed Cardio exercise.` |
| Strength Exercise | Weightlifting, Core training, Indoor climbing | `User performed Strength exercise.` |

## B.5 Zero-Shot Classification

### B.5.1 Task definition for zero-shot classification

We formulate six zero-shot classification tasks related to activity recognition. We compare `SensorLM` with representative LLM baselines across *main activity classification*, *activity-related concept classification*, and *fine-grained recognition*, aiming to evaluate the effectiveness of `SensorLM` in diverse human activity analysis scenarios.

**Main activity classification** encompasses 20 activities. For each activity, the number of samples is indicated in parentheses. These include: *Walking* (881), *Bike* (860), *Playing Sports* (905), *Running* (793), *Aerobics* (910), *Elliptical* (887), *Spinning* (866), *Weightlifting* (842), *Swimming* (882), *Hiking* (848), *Playing Tennis* (821), *CrossFit* (902), *Pilates* (855), *Stairclimber* (863), *Dancing* (852), *Indoor climbing* (854), *Golf* (710), *Skiing* (420), *Snowboarding* (167), and *Kayaking* (212).

For **activity-related concept classification**, we define three sub-tasks. The first one, "*Activity by environmental context*", comprises four classes: *Indoor Sports*, *Outdoor Sports*, *Water Sports*, and *Winter Sports*. The second task, "*Cardio vs. strength training*", distinguishes between two classes: *Cardio exercise* and *Strength exercise*. Similarly, for the third task, "*Locomotion vs. stationary*", we have two classes: *Locomotion exercise* and *Stationary exercise*. The activities included in each concept category are summarized in Table 13.

Finally, we introduce two **fine-grained recognition** tasks. The first task, "*Fine-grained recognition (Gym cardio)*", differentiates among *Elliptical* (887), *Treadmill* (884), *Spinning* (866), and *Stairclimber* (863). The second task, "*Fine-grained recognition (outdoor sports)*", involves classifying among *Hike* (848), *Mountain Bike* (673), *Skiing* (420), and *Snowboarding* (167).

### B.5.2 Implementation for zero-shot classification

**Implementation details for `SensorLM`.** We employ prompt ensembling [25] using a diverse set of rewritten prompts that mirror the structure of pretraining captions. For each label class, we compute the average embedding from the text encoder across all prompts. Zero-shot classification is performed by selecting the label whose average text embedding is most similar to the sensor embedding from the sensor encoder. Examples of the prompts are provided in Table 14; we use a total of 30 prompts.

**Implementation details for LLM baselines.** For zero-shot classification using LLM baselines (i.e., Gemini 2.0 Flash [29] and Gemma-3-27B [30]), we employ a prompt format illustrated in Table 15. The prompt instructs the model to predict a class label based on the provided sensor data, which is formatted as tabular input. Additionally, we also fine-tune Gemini 2.0 Flash [29] on our dataset using supervised activity labels, following a standard supervised fine-tuning (SFT) procedure.

Table 14: **Example of templates used in zero-shot prompt ensemble.**

**Zero-Shot Templates**

```
1. A period of Run was observed during the session.
2. Detected a phase of Run.
3. Data shows Run took place
4. The main action was Run
5. Run was detected during the observed period.
......
```

Table 15: **Zero-shot classification prompt used for LLM baselines.**

**Zero-Shot Classification Prompt**

```
### Overall instruction: Your job is to identify user activity class by analyzing a given day of Fitbit data.
Available Activity Classes: {class_list}

### Sensor Description
##### Heart:
HR: Mean of the instantaneous heart rate in beats per minute. Calculated over a 1 minute window. Unit: Beats/Min
hr_at_rest_mean: Mean of the heart rate in beats per minute during periods of rest. Unit: Beats/Min
hrv_rr_80th_percentile_mean: The 80th percentile of the RR intervals in milliseconds for 5-minute windows with valid
RR intervals. Unit: Msec
hrv_rr_20th_percentile_mean: The 20th percentile of the RR intervals in milliseconds for 5-minute windows with valid
RR intervals. Unit: Msec
hrv_rr_median: The median RR interval in milliseconds for 5-minute windows with valid RR intervals. Unit: Msec
hrv_shannon_entropy_rr: Shannon entropy of the RR intervals for 5-minute windows with valid RR intervals. Unit: Nats
hrv_shannon_entropy_rrd: Shannon entropy of the RR interval differences for 5-minute windows with valid RR
intervals. Unit: Nats
rmssd_percentile_0595: Root mean squared standard deviation of RR intervals in milliseconds for 5-minute windows with
valid RR intervals. Unit: Msec
sdnn_percentile_0595: Standard deviation of RR intervals in milliseconds for 5-minute windows with valid RR
intervals. Unit: Msec
##### Activity:
steps: Number of steps calculated over a 1 minute window. Unit: Steps
jerk_auto: Ratio of lag=1 autocorrelation to energy in 1st 3-axis principal component. Unit: alog_energy
log_energy: Log of sum of 3-axis root mean squared magnitude. Unit: alog_energy
covariance: Estimate of condition number for 3-axis covariance matrix. Unit: acovariance
log_energy_ratio: Log of ratio of sum of energy in 1st 3-axis principal component over energy of 3-axis root mean
squared magnitude. Unit: alog_energy_ratio
zero_crossing_std: Standard deviation of time between zero crossings of 1st 3-axis principal component. Unit: Seconds
zero_crossing_avg: Mean of the time between zero crossings of 1st 3-axis principal component in seconds. Unit: Seconds
axis_mean: Log of the mean square root of the squared X & Z axes of the accelerometer. Unit: a.u.
altim_std: Standard deviation of altimeter readings in Hectopascals. Unit: Hectopascals
kurtosis: Kurtosis of the 3-axis accelerometer root mean squared magnitude. Unit: a.u.
##### Sleep:
sleep_coefficient: Sum of 3-axis max-min range, binned into 16 log-scaled bins. Unit: aSleep
##### EDA:
eda_level_real: Mean tonic skin conductance value in micro Siemens over a 1 minute window. Unit: µSiemens
leads_contact_counts: Number of times the skin conductance sensor electrode leads make contact (likely related to
signal quality). Unit: Counts
ceda_slope_real_micro_siemens: Intraminute slope of SCL values. Unit: µSiemens/Min
skin_temperature_slope: Change in skin temperature in degrees Celsius per minute over a 1 minute window. Unit: °C/Min
wrist_temperatures: Mean skin temperature in degrees Celsius calculated over a 1 minute window. Unit: °C

#### Here is an example output with the predicted activity class and reasoning behind the choice:
{{
  "predicted_class": <predicted_class>,
  "class_probabilities": {{
    "class 1": <probability_of_class_1>,
    "class 2": <probability_of_class_2>,
    "class 3": <probability_of_class_3>,
    "class 4": <probability_of_class_4>,
    ....
    ....
    ....
  }},
  "reasoning": <reasoning>
}}

### Now, it is your turn. Here is a day of Fitbit data for a user in csv format. Each row is a window of 10-minute
average values for the corresponding feature metric. Analyse the Fitbit data thoroughly and make the prediction.
{sensor_data}

### Your output in JSON format containing the predicted class and class probabilities for all classes. The class
probabilities should sum to 1. The predicted class should be from Available Activity Classes list mentioned earlier.
No other classes apart from them should be used. Also give a reason for your prediction:
```

Table 16: **Zero-shot cross-modal retrieval prompt used for LLM baselines.**

**Zero-Shot Cross-Modal Retrieval Prompt**

```
### Overall instruction: You will be given a day of user Fitbit data and a set of 100 captions. Your job is to
analyze the Fitbit data and select top 10 captions that describe the Fitbit data the best.

### Sensor Description
##### Heart:
HR: Mean of the instantaneous heart rate in beats per minute. Calculated over a 1 minute window. Unit: Beats/Min
hr_at_rest_mean: Mean of the heart rate in beats per minute during periods of rest. Unit: Beats/Min
hrv_rr_80th_percentile_mean: The 80th percentile of the RR intervals in milliseconds for 5-minute windows with valid
RR intervals. Unit: Msec
hrv_rr_20th_percentile_mean: The 20th percentile of the RR intervals in milliseconds for 5-minute windows with valid
RR intervals. Unit: Msec
hrv_rr_median: The median RR interval in milliseconds for 5-minute windows with valid RR intervals. Unit: Msec
hrv_shannon_entropy_rr: Shannon entropy of the RR intervals for 5-minute windows with valid RR intervals. Unit: Nats
hrv_shannon_entropy_rrd: Shannon entropy of the RR interval differences for 5-minute windows with valid RR
intervals. Unit: Nats
rmssd_percentile_0595: Root mean squared standard deviation of RR intervals in milliseconds for 5-minute windows with
valid RR intervals. Unit: Msec
sdnn_percentile_0595: Standard deviation of RR intervals in milliseconds for 5-minute windows with valid RR
intervals. Unit: Msec
##### Activity:
steps: Number of steps calculated over a 1 minute window. Unit: Steps
jerk_auto: Ratio of lag=1 autocorrelation to energy in 1st 3-axis principal component. Unit: alog_energy
log_energy: Log of sum of 3-axis root mean squared magnitude. Unit: alog_energy
covariance: Estimate of condition number for 3-axis covariance matrix. Unit: acovariance
log_energy_ratio: Log of ratio of sum of energy in 1st 3-axis principal component over energy of 3-axis root mean
squared magnitude. Unit: alog_energy_ratio
zero_crossing_std: Standard deviation of time between zero crossings of 1st 3-axis principal component. Unit: Seconds
zero_crossing_avg: Mean of the time between zero crossings of 1st 3-axis principal component in seconds. Unit: Seconds
axis_mean: Log of the mean square root of the squared X & Z axes of the accelerometer. Unit: a.u.
altim_std: Standard deviation of altimeter readings in Hectopascals. Unit: Hectopascals
kurtosis: Kurtosis of the 3-axis accelerometer root mean squared magnitude. Unit: a.u.
##### Sleep:
sleep_coefficient: Sum of 3-axis max-min range, binned into 16 log-scaled bins. Unit: aSleep
##### EDA:
eda_level_real: Mean tonic skin conductance value in micro Siemens over a 1 minute window. Unit: µSiemens
leads_contact_counts: Number of times the skin conductance sensor electrode leads make contact (likely related to
signal quality). Unit: Counts
ceda_slope_real_micro_siemens: Intraminute slope of SCL values. Unit: µSiemens/Min
skin_temperature_slope: Change in skin temperature in degrees Celsius per minute over a 1 minute window. Unit: °C/Min
wrist_temperatures: Mean skin temperature in degrees Celsius calculated over a 1 minute window. Unit: °C

#### Here is an example output structure with the predicted top 10 captions:
{{
  "first": <caption_ID>,
  "second": <caption_ID>,
  "third": <caption_ID>,
  "fourth": <caption_ID>,
  "fifth": <caption_ID>,
  "sixth": <caption_ID>,
  "seventh": <caption_ID>,
  "eighth": <caption_ID>,
  "ninth": <caption_ID>,
  "tenth": <caption_ID>
}}

### Now, it is your turn. Here is a day of Fitbit data for a user in csv format. Each row is a window of 10-minute
average values for the corresponding feature metric. Analyze the Fitbit data thoroughly and choose
top 10 captions from the list which best describe the Fitbit data.
{sensor_data}

Here is the set of 100 captions:
{caption_list}

### Your output in JSON format containing the predicted top 10 caption ids:
```

Table 17: **Caption generation prompt used for LLM baselines.**

---

**Caption Generation Prompt**

```
### Overall instruction: You will be given a day of user Fitbit data. Your job is to analyze the Fitbit data and
generate a caption containing semantic information. Semantic information aims to capture the high-level meaning and
context embedded in the sensor data,  essentially interpreting what the individual might be doing or experiencing.

### Sensor Description
##### Heart:
HR: Mean of the instantaneous heart rate in beats per minute. Calculated over a 1 minute window. Unit: Beats/Min
hr_at_rest_mean: Mean of the heart rate in beats per minute during periods of rest. Unit: Beats/Min
hrv_rr_80th_percentile_mean: The 80th percentile of the RR intervals in milliseconds for 5-minute windows with valid
RR intervals. Unit: Msec
hrv_rr_20th_percentile_mean: The 20th percentile of the RR intervals in milliseconds for 5-minute windows with valid
RR intervals. Unit: Msec
hrv_rr_median: The median RR interval in milliseconds for 5-minute windows with valid RR intervals. Unit: Msec
hrv_shannon_entropy_rr: Shannon entropy of the RR intervals for 5-minute windows with valid RR intervals. Unit: Nats
hrv_shannon_entropy_rrd: Shannon entropy of the RR interval differences for 5-minute windows with valid RR
intervals. Unit: Nats
rmssd_percentile_0595: Root mean squared standard deviation of RR intervals in milliseconds for 5-minute windows with
valid RR intervals. Unit: Msec
sdnn_percentile_0595: Standard deviation of RR intervals in milliseconds for 5-minute windows with valid RR
intervals. Unit: Msec
##### Activity:
steps: Number of steps calculated over a 1 minute window. Unit: Steps
jerk_auto: Ratio of lag=1 autocorrelation to energy in 1st 3-axis principal component. Unit: alog_energy
log_energy: Log of sum of 3-axis root mean squared magnitude. Unit: alog_energy
covariance: Estimate of condition number for 3-axis covariance matrix. Unit: acovariance
log_energy_ratio: Log of ratio of sum of energy in 1st 3-axis principal component over energy of 3-axis root mean
squared magnitude. Unit: alog_energy_ratio
zero_crossing_std: Standard deviation of time between zero crossings of 1st 3-axis principal component. Unit: Seconds
zero_crossing_avg: Mean of the time between zero crossings of 1st 3-axis principal component in seconds. Unit: Seconds
axis_mean: Log of the mean square root of the squared X & Z axes of the accelerometer. Unit: a.u.
altim_std: Standard deviation of altimeter readings in Hectopascals. Unit: Hectopascals
kurtosis: Kurtosis of the 3-axis accelerometer root mean squared magnitude. Unit: a.u.
##### Sleep:
sleep_coefficient: Sum of 3-axis max-min range, binned into 16 log-scaled bins. Unit: aSleep
##### EDA:
eda_level_real: Mean tonic skin conductance value in micro Siemens over a 1 minute window. Unit: µSiemens
leads_contact_counts: Number of times the skin conductance sensor electrode leads make contact (likely related to
signal quality). Unit: Counts
ceda_slope_real_micro_siemens: Intraminute slope of SCL values. Unit: µSiemens/Min
skin_temperature_slope: Change in skin temperature in degrees Celsius per minute over a 1 minute window. Unit: °C/Min
wrist_temperatures: Mean skin temperature in degrees Celsius calculated over a 1 minute window. Unit: °C

#### Here are some example captions:

Example 1: Period of Outdoor Bike noted from minute 912 until 917. Outdoor Bike occurred from minute 619 to 628.
Outdoor Bike recorded within the 641-650 minute range. User engaged in Outdoor Bike between minute 928 and 937.
Outdoor Bike took place during the minutes 492 through 502. User engaged in Outdoor Bike between minute 720 and
730. From minute 446 to 456, the user had a period of Outdoor Bike. Outdoor Bike took place during the minutes 572
through 584.

Example 2: Observed Walk spanning minutes 402 to 408. An instance of Walk was identified from minute 652 to 663. A
continuous Walk phase from minute 692 to 703. Detection of Walk activity between minute 1021 and 1035. A Walk period
between minutes 651 and 668. Walk was recorded between minute 785 and 822. A Sleep period between minutes 1346 and
1440.

Example 3: An instance of Walk was identified from minute 279 to 289. An instance of Walk was identified from minute
799 to 828. A Walk period between minutes 279 and 309. Sleep occurred from minute 1303 to 1440.

Example 4: Walk state observed from minute 1132 up to 1139. Outdoor Bike between minutes 1051 and 1064. Outdoor Bike
state observed from minute 489 up to 505. Sleep occurred from minute 4 to 400. The person logged their mood as Excited
at minute 1204. The person registered a mood of Frustrated at minute 773. At minute 773, the individual's feeling is
Frustrated.

### Now, it is your turn. Here is a day of Fitbit data for a user in csv format. Each row is a window of 10-minute
average values for the corresponding feature metric. Analyse the Fitbit data thoroughly and generate the caption which
best describe the Fitbit data.
{sensor_data}

### Your output:
```

## B.6 Few-Shot Learning

For both few-shot learning and linear probing tasks, the sensor encoder of `SensorLM` is frozen. A single linear layer[2] is trained using a multinomial loss with balanced class weights (weights inversely proportional to class frequencies in the training set). This same approach is applied to all SSL baselines for consistency.

For linear probing, we use the full training set (statistics provided in Table 7). For few-shot learning, we randomly sample 5–50 training examples per class and repeat each experiment five times with different random seeds. Table 18 summarizes the class distribution for the "*Anxiety*" and "*Hypertension*" tasks.

Table 18: **Class distribution for the Anxiety and Hypertension prediction tasks.**

| Task / Label | Train | Test |
|---|---|---|
| *Anxiety* | | |
| positive | 55,030 | 34,749 |
| negative | 96,316 | 55,437 |
| *Hypertension* | | |
| positive | 36,349 | 23,353 |
| negative | 114,997 | 66,833 |
| **Total** | 151,346 | 90,186 |

## B.7 Cross-Modal Retrieval

**Implementation details for `SensorLM`.** To evaluate zero-shot cross-modal retrieval, we use top-$k$ recall (R@$k$) for $k \in \{1, 5, 10\}$. This metric measures the proportion of queries for which the ground-truth pairing appears within the top $k$ retrieved results. Retrieval is conducted in both Sensor $\rightarrow$ Text and Text $\rightarrow$ Sensor directions. Cosine similarity between sensor and text embeddings is used as the scoring function to rank candidates. For each query, recall is computed by locating the ground-truth index within the ranked similarity list.

**Implementation details for LLM baselines.** For zero-shot cross-modal retrieval using the LLM baselines (i.e., Gemini 2.0 Flash [29] and Gemma-3-27B [30]), we use a prompt format illustrated in Table 16. The prompt instructs the model to retrieve the top-10 most relevant captions from a set of 100 candidates, based on the provided sensor data formatted as tabular input. We evaluate only in the Sensor $\rightarrow$ Text direction, as Text $\rightarrow$ Sensor retrieval is not feasible due to the context length limitations of the LLMs.

## B.8 Zero-Shot Generalization to Unseen Classes

To assess `SensorLM`'s generalization to novel activities, we conduct a case study involving pretraining on a dataset comprising data only containing 20 activities, followed by testing exclusively on a set of 9 previously unseen classes.

The 20 activities used in the pretraining set include: *Bike*, *Playing Sports*, *Running*, *Aerobics*, *Elliptical*, *Weightlifting*, *Swimming*, *Hiking*, *Playing Tennis*, *CrossFit*, *Core training*, *Pilates*, *Bootcamp*, *Indoor Climbing*, *Golf*, *Kickboxing*, *Skiing*, *Rollerblading*, and *Kayaking*.

The 9 unseen activities evaluated are: *Snowboarding* (167), *Mountain Bike* (673), *Racquetball* (19), *Yoga* (879), *Paddleboarding* (51), *Badminton* (109), *Dancing* (852), *Calisthenics* (61) and *Basketball* (127).

For each unseen activity, we evaluate `SensorLM`'s zero-shot performance using a one-versus-all classification setup. The primary evaluation metric is the Area Under the Receiver Operating Characteristic Curve (AUROC), with $95\%$ bootstrap confidence intervals (CIs) computed using 1000 resampling iterations with replacement [4].

## B.9 Caption Generation

**Implementation details for `SensorLM`.** For sensor captioning, we observe that increasing the number of layers in the multimodal text decoder improves captioning performance. Accordingly, we train a `SensorLM-B` model with 12 decoder layers (compared to the default 3) using ***semantic captions*** during pretraining. During inference, captions are generated autoregressively from sensor data using a `[START]` token.

---

[2] https://scikit-learn.org/stable/modules/generated/sklearn.linear_model.LogisticRegression.html

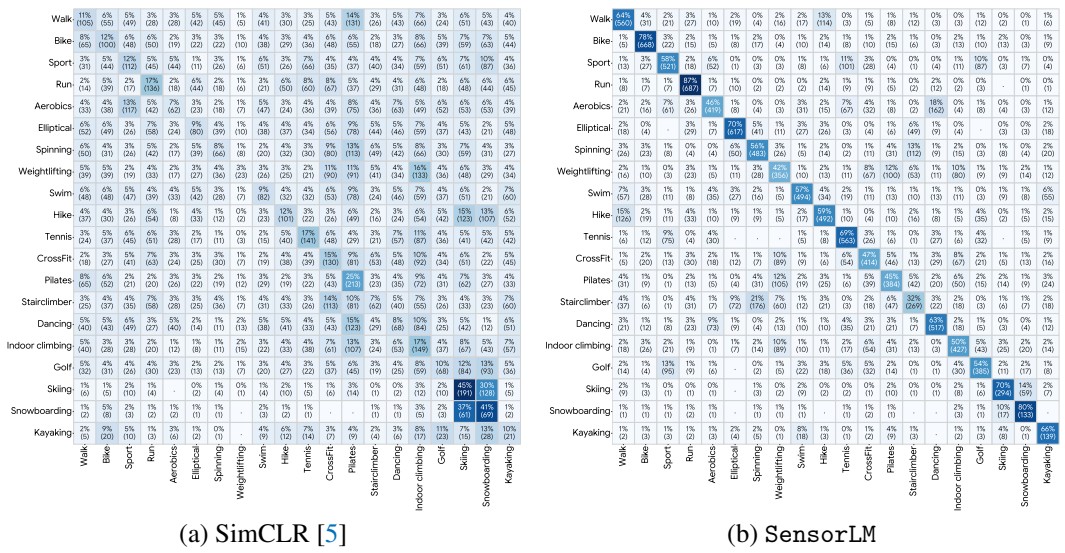

| (a) SimCLR [5] | (b) `SensorLM` |
|---|---|

Figure 9: **Confusion matrices for linear probing activity classification.** Comparison of SimCLR [5] and `SensorLM`-trained sensor encoders on the downstream Activity dataset for 20-class classification. The y-axis represents ground truth labels, and the x-axis shows predicted labels.

Table 19: **Linear probing results.** We use all data for task-specific linear probing. AUROC$^\uparrow$ is used as the metric. Best results of each column are in **bold** and the second best are underlined.

| Task | Hypertension | Anxiety | Activity |
|---|---|---|---|
| SimCLR [5] | 0.56 | 0.64 | 0.68 |
| DINO [3] | 0.56 | 0.62 | 0.66 |
| MSN [2] | 0.58 | **0.66** | 0.72 |
| `SensorLM` | **0.60** | 0.65 | **0.94** |

**Implementation details for LLM baselines.** For caption generation using LLM baselines (i.e., Gemini 2.0 Flash [29] and Gemma-3-27B [30]), we use prompt templates shown in Table 17. These prompts instruct the model to generate a semantic caption based on the provided sensor data, formatted as tabular input.

# C  Additional Results

## C.1  Few-Shot Learning & Linear Probing

Table 19 presents task-specific linear probing results for `SensorLM` compared to several SSL baselines: SimCLR, DINO, and MSN. All evaluations use the full available training data, and performance is measured by AUROC. `SensorLM` achieves competitive or superior performance across all tasks, with the highest AUROC for "*Hypertension*" (0.60) and "*Activity*" (0.94). For "*Anxiety*", `SensorLM` (0.65) performs comparably to MSN (0.66), the top-performing baseline for that task. These results highlight `SensorLM`'s robust representation learning for downstream applications.

Fig. 9 shows confusion matrices for `SensorLM` and SimCLR, illustrating the improved discriminative capability of `SensorLM` embeddings across diverse activity classes.

We also include results on two regression tasks: predicting Age and BMI from one day of sensor data. Using a frozen sensor encoder and a linear regression probe[3], we report Mean Absolute Error

---

[3]https://scikit-learn.org/stable/modules/generated/sklearn.linear_model.LinearRegression.html

Table 20: **Performance of `SensorLM` and baselines on regression tasks.**

| Metrics | Age | | BMI | |
|---|---|---|---|---|
| | MAE$\downarrow$ | MAPE$\downarrow$ | MAE$\downarrow$ | MAPE$\downarrow$ |
| SimCLR | **9.18** | 21.03 | 5.85 | 19.57 |
| DINO | 9.63 | 22.03 | 5.97 | 19.97 |
| MSN | 9.36 | 21.39 | 5.84 | 19.53 |
| `SensorLM` | **9.18** | **20.81** | **5.75** | **19.17** |

Table 21: **Zero-shot cross-modal retrieval.** We evaluate cross-modal retrieval on a held-out dataset containing 40,000 examples, testing performance across varying retrieval set sizes. `SensorLM` achieves consistently strong retrieval performance across all settings. For baseline LLMs, most retrieval tasks are infeasible due to context length limitations (marked as "−").

| Metrics | 100 samples | | | 5,000 samples | | | 40,000 samples | | |
|---|---|---|---|---|---|---|---|---|---|
| | R@1 | R@5 | R@10 | R@1 | R@5 | R@10 | R@1 | R@5 | R@10 |
| *Sensor $\rightarrow$ Text:* | | | | | | | | | |
| Gemma-3-27B [30] | 1.0 | 5.0 | 10.0 | − | − | − | − | − | − |
| Gemini 2.0 [29] | 5.0 | 9.0 | 13.0 | − | − | − | − | − | − |
| `SensorLM` | **100.0** | **100.0** | **100.0** | **98.2** | **99.4** | **99.6** | **96.1** | **98.7** | **99.0** |
| *Text $\rightarrow$ Sensor:* | | | | | | | | | |
| Gemma-3-27B [30] | − | − | − | − | − | − | − | − | − |
| Gemini 2.0 [29] | − | − | − | − | − | − | − | − | − |
| `SensorLM` | **100.0** | **100.0** | **100.0** | **96.7** | **99.2** | **99.4** | **90.0** | **96.9** | **98.1** |

(MAE) and Mean Absolute Percentage Error (MAPE). As shown in Table 20, `SensorLM` improves performance compared to SSL baselines.

## C.2 Cross-Modal Retrieval

Table 21 presents the complete results of zero-shot cross-modal retrieval, comparing `SensorLM` to LLM baselines. Evaluations are conducted on query-target sets of size 100, 5k, and 40k, using Recall@1, Recall@5, and Recall@10 (R@K). These metrics capture the proportion of queries where the correct target appears among the top K retrieved results.

`SensorLM` consistently demonstrates strong retrieval performance across all settings and directions. Notably, it achieves perfect $100\%$ recall at R@1, R@5, and R@10 on the 100-sample benchmark for both Sensor $\rightarrow$ Text and Text $\rightarrow$ Sensor retrieval. Even at the 40k scale, SensorLM maintains high accuracy, with R@1 scores of $96.1\%$ for Sensor $\rightarrow$ Text and $90.0\%$ for Text $\rightarrow$ Sensor.

In contrast, the LLM baselines are largely unable to perform most retrieval tasks due to context length limitations, marked as "−" in the table. Where results are available (mainly for the 100-sample set), their performance is significantly lower than `SensorLM`. These findings highlight `SensorLM` 's superior capability in cross-modal retrieval, where general-purpose LLMs struggle without domain-specific architectural adaptations or training.

In addition to the *semantic* caption example in Fig. 4, Fig. 10 provides qualitative evidence of `SensorLM`'s ability to retrieve accurate *statistical* summaries from raw sensor inputs. `SensorLM` retrieves the ground-truth caption as well as similar alternatives that match sensor statistics (highlighted in yellow), such as means, standard deviations, and extreme values. In contrast, unrelated captions show low similarity scores, reflecting `SensorLM`'s understanding of the statistical structure of the data.

Furthermore, Fig. 11 illustrates `SensorLM`'s capacity for retrieving *structural* descriptions from sensor data. It accurately identifies the ground-truth structural caption and retrieves similar captions

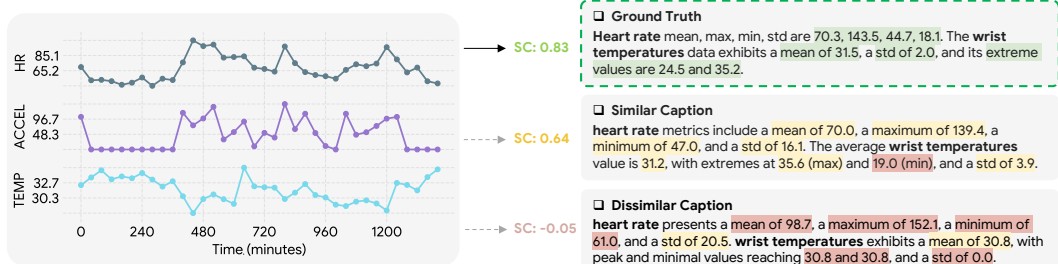

Figure 10: **Zero-shot sensor-to-text retrieval qualitative example for statistical captions.** We show similarity scores (SC) for the correctly retrieved ground truth, a top similar caption, and a dissimilar caption. Green highlights correct statistics; Yellow indicates close matches; and Red denotes distinctly different stasistics. In addition to correctly retrieving the ground truth, SensorLM also demonstrates semantic understanding by assigning high similarity scores to numerically closer candidates.

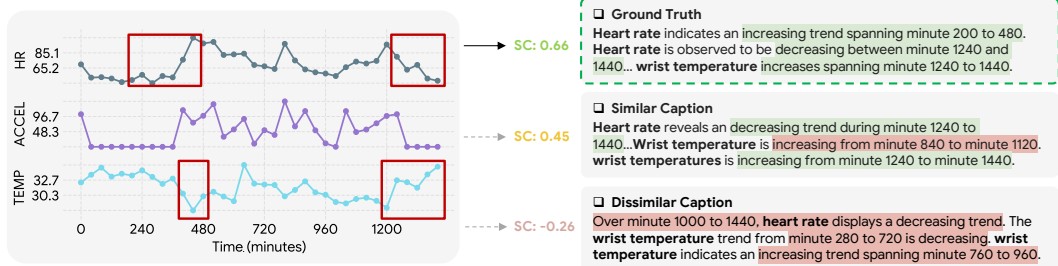

Figure 11: **Zero-shot sensor-to-text retrieval qualitative example for structural captions.** We show similarity scores (SC) for the correctly retrieved ground truth, a top similar caption, and a dissimilar caption. Green highlights correct patterns and time frames; Yellow indicates partial matches; and Red denotes incorrect patterns and time frames. In addition to correctly retrieving the ground truth, SensorLM also demonstrates semantic understanding by assigning high similarity scores to partially correct candidates.

Table 22: **Sensor caption generation results.** We compare SensorLM with LLM baselines using commonly adopted metrics (e.g., BERTScore, METEOR, ROUGE) for natural language generation.

| | BERT Score | | | | |
| Metrics | Precision$^\uparrow$ | Recall$^\uparrow$ | F1$^\uparrow$ | METEOR$^\uparrow$ | ROUGE$^\uparrow$ |
|---|---|---|---|---|---|
| Gemma-3-27B [30] | 0.82 | 0.85 | 0.83 | 0.19 | 0.11 |
| Gemini 2.0 [29] | 0.86 | 0.87 | 0.86 | 0.26 | 0.20 |
| SensorLM | **0.93** | **0.91** | **0.92** | **0.33** | **0.40** |

that describe temporal dynamics, such as trends and spike events. This demonstrates SensorLM 's ability to interpret and express dynamic temporal features within sensor signals.

## C.3 Caption Generation

Table 22 evaluates sensor caption generation performance, comparing SensorLM against LLM baselines. We report standard Natural Language Generation (NLG) metrics: BERTScore (Precision, Recall, F1), METEOR, and ROUGE, which collectively assess semantic similarity, fluency, and content overlap. SensorLM consistently outperforms both LLM baselines across all metrics.

We also present three qualitative examples in Fig. 12, which compare sensor caption outputs from Gemini 2.0 Flash and SensorLM against ground truth references. The examples focus on *semantic caption* generation from wearable sensor inputs. Compared to the baseline, SensorLM demonstrates stronger semantic understanding by generating coherent event descriptions with accurately localized timeframes.

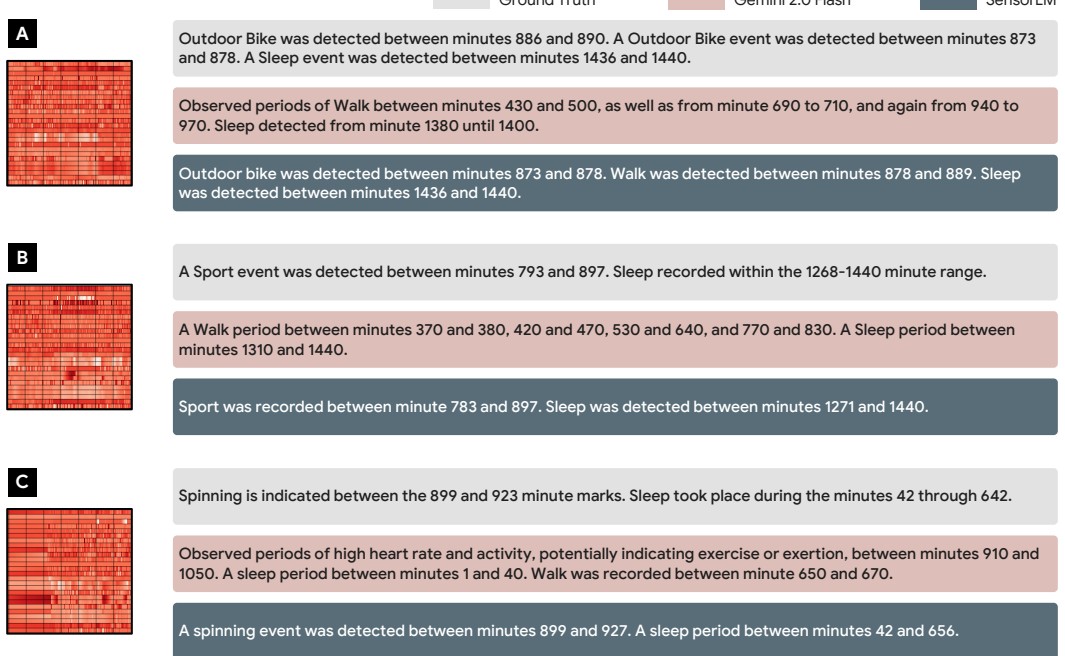

Figure 12: **Qualitative comparison of sensor caption generation between Gemini 2.0 Flash and** `SensorLM` **against ground truth.** We show various examples of *semantic caption* generation from wearable sensor inputs. Compared to the baseline, `SensorLM` demonstrates stronger semantic understanding by generating coherent events with near-correct localized timeframes.

Table 23: **Comparisons of** `SensorLM` **caption versions.** We evaluate different combinations of caption types used during pretraining. $AUROC^{\uparrow}$ is used as the metric. Best results of each column are in **bold** and the second best are underlined. Default setting used in the main experiments is marked in gray.

| Caption Variant | Zero-Shot Activity | Linear Probing Activity | Linear Probing Anxiety | Linear Probing Hypertension | Retrieval (40k) Recall@1 |
|---|---|---|---|---|---|
| statistical | 0.51 | 0.76 | 0.67 | **0.63** | **1.00** |
| structural | 0.50 | 0.78 | 0.63 | 0.59 | **1.00** |
| semantic | 0.71 | **0.95** | 0.65 | 0.60 | 0.89 |
| statistical + semantic | 0.66 | 0.84 | **0.68** | 0.62 | **1.00** |
| structural + semantic | **0.84** | 0.94 | 0.65 | 0.60 | **1.00** |
| statistical + structural | 0.49 | 0.79 | 0.67 | **0.63** | 0.64 |
| statistical + structural + semantic | 0.66 | 0.86 | **0.68** | **0.63** | 0.90 |

## C.4 Complete Ablation Results

Here we present the complete results for the ablation studies introduced in Sec. 5.3. Table 23 examines how different combinations of sensor caption types (*statistical*, *structural*, and *semantic*) affect `SensorLM`'s performance across downstream tasks. Semantic captions are critical for zero-shot activity recognition, while combining them with structural captions further enhances performance across tasks, including cross-modal retrieval. The results also reveal trade-offs: statistical captions improve performance on "*Anxiety*" and "*Hypertension*", but slightly reduce accuracy on "*Activity*".

Table 24 evaluates different architectural variants of `SensorLM` during pretraining. `SensorLM (CoCa)` consistently outperforms single-objective variants (`SensorLM (CLIP)` and `SensorLM (Cap)`) across key metrics for both zero-shot classification and linear probing, validating the benefits of integrating contrastive and generative objectives.

Table 24: **Comparisons of `SensorLM` architectural variants.** We compare different choices used during pretraining. Default setting used in the main experiments is marked in gray.

| Metrics | Zero-Shot | | | Linear Probing | | |
|---|---|---|---|---|---|---|
| | AUROC$^\uparrow$ | F1$^\uparrow$ | Balanced Acc.$^\uparrow$ | AUROC$^\uparrow$ | F1$^\uparrow$ | Balanced Acc.$^\uparrow$ |
| `SensorLM (CLIP)` | 0.83 | **0.29** | 0.31 | 0.93 | 0.53 | 0.55 |
| `SensorLM (SigLIP)` | 0.78 | 0.24 | 0.26 | 0.87 | 0.38 | 0.40 |
| `SensorLM (Cap)` | 0.55 | 0.01 | 0.05 | 0.90 | 0.32 | 0.52 |
| `SensorLM (CoCa)` | **0.84** | **0.29** | **0.32** | **0.94** | **0.57** | **0.60** |

Table 25: **Comparisons between UniMTS and `SensorLM`.** Performance on zero-shot and linear probing (LP) settings across activity recognition and metabolic health tasks.

| Model | Zero-Shot | | | Linear Probing (LP) | | |
|---|---|---|---|---|---|---|
| | Activity (20-class) | Gym Cardio | Outdoor Sports | Activity | Anxiety | Hypertension |
| UniMTS [44] | 0.71 | 0.73 | 0.65 | **0.94** | 0.65 | 0.58 |
| `SensorLM` | **0.84** | **0.76** | **0.86** | **0.94** | **0.65** | **0.60** |

## C.5 Comparisons to Existing Sensor Foundation Models

UniMTS [44] proposes a unified pre-training method for motion time series that synthesizes sensor data from human skeletons and uses contrastive learning to align it with natural language. To compare with UniMTS, we implement their captioning and pretraining method on our sensor-text data as a zero-shot baseline. Specifically, it employs a CLIP [25] objective and treats activity descriptions as language input, corresponding to the *semantic* caption in our framework. We adapt and train UniMTS under our experimental setup and compare with our results in Table 25.

The table confirms that `SensorLM` outperforms UniMTS across all zero-shot evaluation tasks. To further understand the effect of hierarchical captions on this existing work, we observe from the table that: (1) Adding structural captions improves zero-shot AR performance, aligning with our caption ablation findings. (2) Including statistical captions improves metabolic prediction tasks, though with a slight trade-off in AR performance, again consistent with trends discussed previously. These results support both the quality of the dataset and the value of the hierarchical captioning strategy in improving generalization across diverse tasks.

## C.6 Exploration of Loss Weights

As our work introduces a modular and generic pretraining framework, it is straightforward to attach, remove, or reweight objectives. This modularity allows systematic ablations, rapid integration of new components, and reveals design tradeoffs unexplored in prior image-text studies. Here we sweep the relative weights of the contrastive ($\lambda_{con}$) and captioning ($\lambda_{cap}$) losses beyond fixed settings by CLIP ($\lambda_{con} = 0$) and CoCa ($\lambda_{con} = \lambda_{cap} = 1$). We find clear task-specific trade-offs: a $1 : 2$ balance improves zero-shot activity recognition, whereas $5 : 1$ yields the best metabolic-health prediction. These results indicate that careful tuning, enabled by our framework, yields measurable gains.

## C.7 Robustness to OOD Data and Missingness

Here we stratify our testing set to study the performance of `SensorLM` on data from OOD devices and with missingness and missing modalities. All reported results in the main paper already include these OOD and missing-data cases.

### C.7.1 Out-of-Distribution Devices

Our pretraining set contains **18** smartwatch and fitness-tracker models, while the metabolic downstream dataset has an additional **17** device types. As a result, we are able to demonstrate generalizabil-

Table 26: **Exploration of the loss weights in pretraining.** We sweep the relative weights of the contrastive ($\lambda_{con}$) and captioning ($\lambda_{cap}$) losses beyond the fixed CLIP ($\lambda_{con} = 0$) and CoCa ($\lambda_{con} = \lambda_{cap} = 1$) settings, establishing different tradeoffs.

| Weights | | Zero-Shot | | | Linear Probing | | |
|---|---|---|---|---|---|---|---|
| $\lambda_{con}$ | $\lambda_{cap}$ | Activity (20-class) | Gym Cardio | Outdoor Sports | Activity | Anxiety | Hypertension |
| 1 | 0.2 | 0.82 | 0.77 | 0.82 | 0.92 | **0.66** | **0.62** |
| 1 | 1 | 0.84 | 0.76 | **0.86** | **0.94** | 0.65 | 0.60 |
| 1 | 2 | **0.85** | **0.78** | 0.85 | 0.93 | 0.65 | 0.60 |

Table 27: **Metabolic health prediction results on in-distribution (ID) and out-of-distributuion (OOD) devices.** AUROC$^{\uparrow}$ is used for classification tasks (Anxiety, Hypertension) and MAE$^{\downarrow}$ for regression tasks (Age, BMI).

| Split | Anxiety (AUROC$^{\uparrow}$) | Hypertension (AUROC$^{\uparrow}$) | Age (MAE$^{\downarrow}$) | BMI (MAE$^{\downarrow}$) |
|---|---|---|---|---|
| ID | 0.65 | 0.61 | 9.24 | 5.74 |
| OOD | 0.72 | 0.55 | 8.55 | 5.79 |

Table 28: **Effect of missingness on `SensorLM` performance.** We report AUROC$^{\uparrow}$ across different missingness levels and sensor modality settings.

| *Varying missing percentage* | | | |
|---|---|---|---|
| **Missing %** | **Activity** | **Anxiety** | **Hypertension** |
| <40% | 0.94 | 0.68 | 0.62 |
| 40–60% | 0.94 | 0.65 | 0.59 |
| 60–80% | 0.91 | 0.65 | 0.65 |
| *Specific sensor modality missing* | | | |
| **Modality Missing** | **Activity** | **Anxiety** | **Hypertension** |
| w/o missing | 0.94 | 0.67 | 0.62 |
| w/ missing | 0.89 | 0.65 | 0.60 |

ity on the metabolic test set by comparing performance on in-domain (N=83,243) and out-of-domain (N=6,943) devices. We follow the setting in the main paper (Sec. 5.1) and report classification (AUROC) and regression (MAE) results. As the table confirms, the model exhibits sound OOD generalization: while "BMI" and "Hypertension" predictions show slight variation, "Anxiety" and "Age" predictions may improve under OOD conditions.

### C.7.2  Missingness

In both our pretraining and downstream datasets, sensor missingness arises naturally (e.g., devices like *Fitbit Versa* or *Charge* lack EDA sensors). We perform thorough preprocessing for data missingness following established practices: When an entire sensor modality is absent, we impute values with the population mean computed from the pretraining data. For temporal gaps within a sensor stream (e.g., dropouts), we apply linear interpolation across gaps and forward/backward filling at sequence boundaries. Consequently, `SensorLM` is trained on data with up to 80% missingness. It learns to model missingness patterns and shows robustness across scenarios.

To further investigate the performance under missingness, we conducted additional experiments on (1) subgroup analysis stratified by percentage of missingness, and (2) comparisons when certain sensor modalities are entirely absent. First, we observe only mild degradation as missingness increases from <40% to 60–80%. Second, when specific sensor modalities (e.g., EDA or HRV) are entirely absent,

we note a moderate AUROC decrease (0.02–0.09) relative to the full-modality setting. These findings confirm that, although performance degrades slightly under missingness, `SensorLM` remains robust and flexible with incomplete sensor data.

## D  Societal Impact

**Broader Impacts.** Ubiquitous health technologies, particularly wearables, offer tremendous potential to transform healthcare through continuous, longitudinal personal monitoring. However, the complexity of raw sensor data often restricts insights to low-level metrics. `SensorLM` addresses this challenge by translating multimodal sensor signals into natural language, enabling a more intuitive and accessible interface for both consumers and domain experts. This capability promotes clearer, more actionable insights, potentially fostering more proactive, personalized, and preventative health management.

**Limitations and Ethical Considerations.** While consumer health research holds great promise, it must be guided by careful attention to safety, fairness, and privacy. The possibility of misuse by malicious actors underscores the importance of responsible development. Although we advocate for open science, health data–by its nature–requires a delicate balance between research reproducibility and participant confidentiality.

Importantly, `SensorLM` is a research prototype and is not intended for clinical use. It has not been validated as a diagnostic tool and should not be used for medical decision-making without formal regulatory approval. Clinical deployment would necessitate rigorous validation and compliance with relevant healthcare regulations.

Finally, while `SensorLM`'s methodology is designed to be generalizable, our current evaluation is limited to specific wearable devices and sensor modalities. Further research is needed to assess its performance across broader device ecosystems, data modalities, and population groups.

