# OpenReview forum: "SensorLM: Learning the Language of Wearable Sensors"
_NeurIPS.cc/2025/Conference — NeurIPS 2025 poster_

### Official Review · Reviewer_ZU6J · 2025-06-29

**Clarity:** 3
**Significance:** 4
**Originality:** 4
**Rating:** 5
**Confidence:** 3

**Summary:**

This paper introduces SensorLM, a family of sensor-language foundation models that learn from large-scale wearable sensor data paired with automatically generated textual descriptions. The authors propose a hierarchical captioning pipeline (statistical, structural, semantic) to describe over 59 million hours of multimodal data collected from more than 100,000 individuals, enabling alignment with natural language for downstream tasks. The model integrates contrastive and generative objectives, extending prior multimodal architectures (e.g., CLIP, CoCa) to the sensor-text domain. Extensive evaluations demonstrate SensorLM's superiority in zero-shot recognition, few-shot learning, and cross-modal retrieval, making it a compelling step toward interpretable and generalizable sensor understanding.

**Questions:**

1. Have the authors evaluated SensorLM’s generalization ability on non-wearable time-series data, such as medical monitoring signals or industrial sensors?
2. What is the computational overhead of SensorLM at inference time compared to conventional sensor models, especially in edge or real-time environments?
3. Can the hierarchical captioning pipeline be adapted or extended to support human-written descriptions or other forms of supervision beyond rule-based heuristics?
4. How robust is the model when some sensor modalities are missing, noisy, or partially corrupted at inference time?

**Ethical Concerns:**

["NO or VERY MINOR ethics concerns only"]

**Limitations:**

Please see above.

**Quality:**

4

**Strengths And Weaknesses:**

### [Strengths]
1. The hierarchical captioning pipeline—comprising statistical, structural, and semantic levels—offers a unique way to supervise time-series signals with interpretable textual labels.
2. SensorLM demonstrates strong empirical results across a wide range of datasets, tasks, and architectures, showing robustness and generalization beyond conventional sensor models.
3. The method supports test-time inference without requiring fine-tuning or task-specific labels, which is particularly valuable for real-world deployment in personalized or low-resource settings.
4. The authors provide a comprehensive evaluation that includes classification, retrieval, and metabolic prediction tasks, highlighting the model's versatility across modalities and objectives.

### [Weaknesses]
1. The quality of the pretraining data relies on heuristically generated captions, which may introduce noise or bias that affects the downstream alignment performance.
2. The model is currently evaluated only on wearable sensor data, and its applicability to other time-series domains—such as industrial or medical sensors—remains untested.

---

> ### Author Rebuttal · Authors · 2025-07-31
>
> Thank you very much for acknowledging the novelty and the contributions of our work! We are glad that you found the paper comprehensive and valuable with strong empirical results. In the following, we address your concerns in detail. We hope that these will further clarify our work and lead to a favorable increase of the score.
>
> &nbsp;
>
> > *W1: The quality of the pretraining data relies on heuristically generated captions, which may introduce noise or bias that affects the downstream alignment performance.*
>
> We appreciate the reviewer’s concern about potential noise or bias introduced by rule‑based captions, particularly for the statistical and structural components. The semantic captions, however, are derived from *self‑reported* user events. Our hierarchical scheme therefore blends human‑provided information with *programmatically* extracted patterns, reducing reliance on **any single heuristic**. In addition, every caption is paraphrased through diverse templates generated by LLMs (**Appendix A.2**), which broadens linguistic variety and better reflects how users might describe sensor traces. This strategy delivers **scalability** that manual annotation cannot match, allowing us to assemble a large-scale sensor-language corpus.
>
> Empirically, our results support that SensorLM remains **robust** despite possible noise, achieving strong alignment performance across diverse downstream tasks. **Section 5.3** and **Appendix C.5** analyze the contribution of each caption type and training strategy, offering further insight into alignment dynamics.
>
> We believe this trade-off between **caption realism** and **scalability** is a practical and necessary step toward enabling large-scale sensor-language learning. In future work, incorporating more naturalistic language data (e.g., user descriptions or clinical notes) could further improve alignment quality and enable more model capabilities. We will add more discussion on the trade-off of rule-based captioning and human annotations in the revised manuscript.
>
> &nbsp;
>
> > *W2 & Q1: The model is currently evaluated only on wearable sensor data, and its applicability to other time-series domains—such as industrial or medical sensors—remains untested.*
>
> Thank you for the thoughtful comment. While our current evaluation focuses on wearable sensor data, the SensorLM framework is designed to be **modality-agnostic** and generalizable to other time-series domains. The captioning pipeline for deriving statistical and structural patterns from raw signals, and the ability to incorporate new semantic information via captions, is **not specific** to wearable devices. Our **pretraining objectives** and **modular architecture** can also be readily adapted to structured inputs from industrial or medical sensors. We see this as a promising direction for future work and believe that SensorLM lays the foundation for scalable sensor-language learning across diverse domains. We will add these discussions in the revised manuscript.
>
> &nbsp;
>
> > *Q2: What is the computational overhead of SensorLM at inference time compared to conventional sensor models, especially in edge or real-time environments?*
>
> SensorLM demonstrates competitive inference time overhead compared to conventional baselines. For zero-shot inference, the combined sensor and text encoders require **122 GFLOPs**. In few-shot or linear probing scenarios, SensorLM's sensor encoder, at **92 GFLOPs**, offers comparable efficiency to the SSL baselines (**105 GFLOPs**), maintaining a total cost around **100 GFLOPs**. The complete caption generation process requires **155 GFLOPs**; however, this cost is justified by the model's distinctive interpretive capabilities.
>
> Challenges for deployment in highly constrained environments are similar to those of baseline SSL methods, but these can be mitigated through established optimization strategies like model compression or hybrid on-device/cloud inference.
>
> We will include a detailed discussion of these considerations in the revised manuscript.
>
> &nbsp;
>
> > *Q3: Can the hierarchical captioning pipeline be adapted or extended to support human-written descriptions or other forms of supervision beyond rule-based heuristics?*
>
> Yes, the hierarchical captioning pipeline is modular and can be adapted to incorporate human-written descriptions, clinical notes, or expert annotations in place of or alongside existing captions. Its design supports seamless integration of diverse textual inputs. While large-scale expert annotations are challenging to obtain, our caption generation pipeline and pretraining framework is readily adaptable should such supervised data become available. As a first step, our rule-based approach makes scalable sensor-language pretraining possible.
>
> We will add a discussion section on the adaptability to human-written or other forms of supervision in the revised manuscript.
>
> &nbsp;
>
> > *Q4: How robust is the model when some sensor modalities are missing, noisy, or partially corrupted at inference time?*
>
> Thank you for raising this point. Our model is trained to be robust to missing or partially corrupted modalities. During pretraining we encounter natural sensor missingness and apply imputation strategies following established practice  [ref 6, ref 20], which allows the model to learn such patterns.
>
> To further understand the effect of missingness on the performance, we conduct additional experiments on (1) *subgroup analysis stratified by percentage of missingness*, and (2) *comparisons when certain sensor modalities are entirely absent*.
>
> |Missing Ratio|Activity|Anxiety|Hypertension|
> |-|-|-|-|
> |<40%|0.94|0.68|0.62|
> |40-60%|0.94|0.65|0.59|
> |60-80%|0.91|0.65|0.65|
>
> |Modality Missing|Activity|Anxiety|Hypertension|
> |-|-|-|-|
> |w/o missing|0.94|0.67|0.62|
> |w missing|0.89|0.65|0.60|
>
> As shown in the tables above, while missing modalities and increased missingness ratio lead to slight performance degradation, the model maintains strong robustness and generalization across tasks.
>
> For noisy or corrupted inputs we likewise anticipate some degradation, as is typical for most models; however, consistent with the empirical scaling laws reported in **Appendix C.1**, SensorLM benefits from large-scale, diverse training data that may help mitigate this effect.
>
> We will report these results and add a robustness discussion in the revised manuscript.
>
> &nbsp;
>
> ---
> We hope the above responses have addressed all concerns from the reviewer adequately. We would also like to update the reviewer on our commitment to open‑sourcing SensorLM:
> - **Dataset**. Under our IRB‑approved protocol, we have obtained informed consent to release a **de‑identified version** of the downstream dataset, including hypertension, anxiety, age, and BMI labels. Access will be limited to researchers with verified academic affiliations.
> - **Pretrained model**. We will release **SensorLM checkpoints** trained on these data together with example code for inference and fine-tuning.
> - **Code**. We will open-source the complete caption-generation pipeline to facilitate reuse by the broader research community.
>
> Once again, thanks for your time and constructive feedback. We would really appreciate it if the reviewer could consider raising their scores and championing the paper.

---

> > ### Author Response · Authors · 2025-08-05
> >
> > We would like to thank the reviewer again for your time and effort. We would greatly appreciate the opportunity to engage with you, as we have put a significant effort into providing a detailed response to your questions, including new experiments on **computation overhead** and **robustness to imperfect & OOD data**. We believe the additional results and clarifications should have successfully addressed all your concerns.
> >
> > Given there is only **2 days** left in the discussion period, we wanted to double check whether there were any final clarifications the reviewer would like. If the concerns are clarified and the reviewer is convinced of the novelty and completeness of our work, we'd be grateful if the reviewer could update your review and score to reflect that, so we know that our response has been seen. Once again, thanks for your time and dedication to the review process.

---

> > ### Comment · Reviewer_ZU6J · 2025-08-06
> >
> > I appreciate the detailed feedback from the authors, which addressed some concerns. Good luck.

---

> > > ### Author Response · Authors · 2025-08-06
> > >
> > > Thank you for your positive remark! We are glad that our responses have addressed your concerns. We are happy to address any further questions. Given all concerns have been clarified, we would be grateful if the reviewer could consider updating your review and score to reflect them.

---

### Official Review · Reviewer_oLSX · 2025-07-01

**Clarity:** 3
**Significance:** 3
**Originality:** 2
**Rating:** 3
**Confidence:** 4

**Summary:**

This paper presents a new sensor data captioning strategy, which has three hierarchies: statistical, structural, and semantic captions. Then, the paper presents SensorLM, aiming to generate captions based on sensor data. SensorLM is a hybrid method combining contrastive alignment and text generation. Experiments demonstrate that SensorLM is superior to baselines methods on diverse downstream tasks.

**Questions:**

What does it mean by "zero-shot" in your context? Does it mean the test classes are unseen, or the test datasets are collected separately from your training set?

**Ethical Concerns:**

["NO or VERY MINOR ethics concerns only"]

**Final Justification:**

The paper has a good structure and clear writing. The idea is interesting and meaningful, especially the dataset construction part. However, the SensorLM model itself has minor originality, and the experimental studies are specific to the proposed model and datasets, lacking general insights and guidance to the general domain. Thus, while also fine for the research track, I would advocate this paper more for presenting at the dataset track. Furthermore, the reproducibility of this paper is inadequate. Thus, I am on the borderline of this paper.

**Limitations:**

Yes

**Paper Formatting Concerns:**

No formatting concerns.

**Quality:**

2

**Strengths And Weaknesses:**

Strengths:
1. The paper is overall well written and organized. It is easy to follow the idea.
2. Building large datasets is a fundamental part of the community and would largely inspire the following research.
3. The idea of hierarchical captioning is interesting and meaningful.

Weaknesses:
1. The main contribution of this work is the sensor-text dataset with hierarchical captions. However, the authors do not publish the dataset. Although the data collection details are presented. Datasets are different from models, which are hard and time-consuming to reproduce. Thus, it is critical to publish the dataset.
2. The quality of the dataset is not studied. It would be better if the authors could show how existing studies work on the new dataset and how the hierarchical captions impact the performance.
3. The novelty of the SensorLM model is limited. It combines generative and contrastive learning, which is not new. In addition, the authors use $\lambda_{con}=1$ and $\lambda_{cap}=1$ as the default setting, which is CoCa actually. The paper fails to show the benefits of tuning the loss weights $\lambda$.
4. In Tables 2 and 3, the results of the baselines are nearly random. It is quite unfair to make such a comparison. The two baselines should be fine-tuned using the proposed dataset first, then make inferences and comparisons.
5. The paper would be more suitable for the dataset track instead of the research track.

---

> ### Author Rebuttal · Authors · 2025-07-31
>
> Thank you for your thoughtful feedback, your comments have helped us strengthen the evidence for our approach. However, we believe that there are several **important misunderstandings** which we would like to clarify and address point-to-point:
>
> &nbsp;
>
> > *W1: The main contribution of this work is… it is critical to publish the dataset.*
> >
> > *W5: The paper would be more suitable for the dataset track instead of the research track.*
>
> We respectfully disagree with this assessment. We would like to clarify that SensorLM’s key contribution **extends well beyond a “sensor‑text dataset”**, addressing the **full pipeline** of sensor‑language pretraining with several advances:
>
> 1. **Defining paired language for sensor data where none naturally exist**. We introduce a *scalable* hierarchical caption framework, enabling alignment learning at scale, resulting in the *largest* sensor‑language corpus to date, orders of magnitude larger than prior studies (Table 1).
> 2. **Determining what makes for effective captions**. Contrary to the common assumption [ref 36], we show that relying solely on high‑level semantic descriptions is *sub‑optimal* across tasks (see **W2** responses).
> 3. **Establishing empirical scaling laws for sensor‑language alignment**. We conduct the *first* large‑scale study of sensor-language alignment and demonstrate clear scaling laws (also noted by Reviewer wWho).
> 4. **Providing a *modular and generic* pre‑training framework**. The framework makes it straightforward to *attach, remove, or re‑weight* objectives. This *modularity* allows systematic ablations, fast integration of new components, and reveals design trade‑offs unexplored in prior image-text studies (see **W3** responses).
>
> These methodological, empirical, and conceptual advances go well beyond dataset construction and align with the expectations of the main track. Importantly, they align with the **NeurIPS reviewer guideline** that
>
> > *”originality does not necessarily require introducing an entirely new method but **can come from new insights, novel combinations, or improved understanding of existing approaches**”*
>
> &nbsp;
>
> **Open‑source commitment.** In addition, we would also like to update the reviewer on our commitment to open‑sourcing SensorLM:
> - **Dataset**. Under our IRB‑approved protocol, we have obtained informed consent to release a **de‑identified version** of the downstream dataset (including hypertension, anxiety, age, and BMI labels) to researchers with verified academic affiliations.
> - **Pretrained model**. We will release **SensorLM checkpoints** trained on these data with example code for inference and fine-tuning.
> - **Code**. We will open-source the complete caption-generation pipeline to benefit the broader research community.
>
> We hope these clarified contributions, along with our open-sourcing commitment, encourage the reviewer to reassess our paper’s novelty and significance.
>
> &nbsp;
>
> > *W2: The quality of the dataset is not studied…how the hierarchical captions impact the performance.*
>
>
> Thanks for the suggestion. We would like to clarify that the dataset quality **has been** carefully benchmarked with established SSL methods.
>
> First, **SSL baselines** benchmarked under the few-shot protocol in **Sec. 5.1** (Appendix **Table 19, Fig. 9**) show that models pretrained with SensorLM and hierarchical captions consistently surpass SSL baselines trained on the same data.
>
> In addition, we adapt **UniMTS** [ref 36], a foundation model capable of zero-shot classification by using a CLIP objective and *semantic* activity caption, to our dataset (row 1 below) and examine how our hierarchical captioning strategies (rows 2-3 below) affect performance:
>
> | |Caption|Zero-Shot|||Linear Probing|||
> |-|-|-|-|-|-|-|-|
> | | |20-class Activity|gym cardio|outdoor sports|Activity|Anxiety|Hypertension|
> |UniMTS|semantic|0.71|0.73|0.65|**0.94**|0.65|0.58|
> |UniMTS + our caption|semantic + structural|0.83|0.75|0.84|0.92|0.66|0.62|
> |UniMTS + our caption|semantic + structural + statistical|0.67|0.63|0.76|0.83|**0.67**|**0.63**|
> |SensorLM|semantic + structural|**0.84**|**0.76**|**0.85**|**0.94**|0.65|0.60|
>
> The table shows that adding **structural** captions improves zero‑shot activity recognition, whereas **statistical** captions benefits metabolic‑health tasks, consistent with our ablation study (**Table 4, Sec. 5.3**). These findings confirm both dataset quality and the effectiveness of the hierarchical caption strategy.
>
> We will incorporate the additional baseline and analyses into the revised manuscript.
>
> &nbsp;
>
> > *W3: The novelty of the SensorLM model…benefits of tuning the loss weights $\lambda$.*
>
> We respectfully disagree with the claim. Our contribution extends **beyond architectural design** (see our responses to **W1**) and addresses the **full pipeline** of sensor‑language pretraining, offering new insights into aligning wearables with natural language.
>
> While our architecture skeleton follows established image-text paradigms, our work introduces a *modular* and *generic* pretraining framework that makes it straightforward to *attach, remove, or re‑weight* objectives. This **modularity** allows systematic ablations, rapid integration of new components, and reveals design tradeoffs unexplored in prior image-text studies.
>
> To support this claim, we study two extensions that would be less practical without such modular design.
>
> **(1) Loss-weight exploration**. Per reviewer suggestion, we sweep the relative weights of the contrastive ($\lambda_{con}$) and captioning ($\lambda_{cap}$) losses beyond fixed settings by CLIP ($\lambda_{con}=0$) and CoCa ($\lambda_{con}=\lambda_{cap}=1$). We find clear task‑specific trade‑offs: a 1:2 ratio boosts zero‑shot activity recognition, whereas 5:1 yields the best metabolic‑health prediction. These results show that careful tuning, enabled by our framework, produces measurable gains.
>
> |$\lambda_{con}$|$\lambda_{cap}$|Zero-Shot|||Linear Probing|||
> |-|-|-|-|-|-|-|-|
> |||**Activity (20-class)**|**gym cardio**|**outdoor sports**|**Activity**|**Anxiety**|**Hypertension**|
> |1|0.2|0.82|0.77|0.82|0.92|**0.66**|**0.62**|
> |1|1|0.84|0.76|**0.86**|**0.94**|0.65|0.60|
> |1|2|**0.85**|**0.78**|0.85|0.93|0.65|0.60|
>
> **(2) Sensor-reconstruction head.**  We append either an auto‑encoder (AE) or vector‑quantized auto‑encoder (VQ‑AE) decoder to the sensor encoder, so that training jointly optimizes *contrastive alignment*, *captioning*, and *signal reconstruction*​. Adding the reconstruction loss improves health‑prediction metrics with only a modest drop in activity recognition. The VQ‑AE variant achieves the best overall trade‑off:
>
> | |Zero-Shot|||Linear Probing|||
> |-|-|-|-|-|-|-|
> ||**Activity (20-class)**|**gym cardio**|**outdoor sports**|**Activity**|**Anxiety**|**Hypertension**|
> |Con + Cap|**0.84**|**0.76**|**0.86**|**0.94**|0.65|0.60|
> |Con + Cap + AE|0.76|0.67|0.82|0.89|0.68|0.63|
> |Con + Cap + VQ-AE|0.77|0.68|0.74|0.91|**0.69**|**0.65**|
>
> These extensions, enabled by the modular framework, show that it supports rapid experimentation and yields concrete performance benefits. Again, this is consistent with the **NeurIPS reviewer guideline**:
>
> > *”originality does not necessarily require introducing an entirely new method when the work enables new insights or capabilities"*
>
> We hope the clarifications and new results clarify the significance of our contribution. We will incorporate the discussions and results into the manuscript.
>
> &nbsp;
>
> > *W4: In Tables 2 and 3, the results of the baselines are nearly random… fine-tuned using the proposed dataset first, then make inferences and comparisons.*
>
> Thank you for the comment. We acknowledge that the LLM baselines in **Tables 2 and 3** were not originally designed for high-dimensional time-series reasoning.
>
> To address the reviewer’s concern, we fine-tuned **Gemini 2.0 Flash** on our dataset using **supervised activity labels**, following a standard supervised fine-tuning (SFT) procedure. We show zero‑shot balanced accuracy under the same protocol as Table 2:
>
> | |20-class|Environment|Cardio / strength|Locomotion / stationary|Gym cardio|Outdoor sports|
> |-|-|-|-|-|-|-|
> |Gemini 2.0|0.07|0.25|0.52|0.55|0.26|0.29|
> |Gemini 2.0 (SFT)|0.1|0.28|0.53|0.55|0.35|0.32|
> |SensorLM-XL|**0.31**|**0.38**|**0.66**|**0.58**|**0.51**|**0.53**|
>
> The table confirms that fine‑tuning yields only *marginal* gains over the vanilla Gemini, and still falls well short of SensorLM, despite Gemini 2.0 Flash being substantially larger.
>
> These results highlight two persistent limitations of (fine‑tuned) LLMs:
> 1. Due to intrinsic context‑length constraints, LLM requires downsampling to interpret raw sensor data, limiting their ability to capture fine‑grained dynamics;
> 2. SFT on supervised activity labels reduces instruction‑following ability of LLMs in other tasks, leading to limited zero‑shot generalization.
>
> We will add the fine‑tuned results in the manuscript and include **UniMTS** (see our response to **W2**) as a stronger zero-shot baseline for a more thorough comparison.
>
> &nbsp;
>
> > *Q1: What does it mean by "zero-shot" in your context?...collected separately from your training set?*
>
> Thanks for the question. By “zero‑shot” we adopt the convention used in CLIP and CoCa: the model is evaluated exactly as trained, with **no** task‑specific tuning or calibration. Our test cohorts are participant‑independent and entirely **disjoint** from the pretraining pool.
>
> Additionally, for categorical tasks (**Table 2**) and unseen activity classes (**Fig. 6**), we present label names **never** observed during pretraining; the model must map them to sensor patterns through the learnt sensor-language representation.
>
> &nbsp;
>
> ---
> We hope these clarifications and our open-sourcing commitment encourage the reviewer to reassess the contributions of the paper. We are more than happy to answer more questions. If the discussions and results successfully addressed all your concerns, please consider raising your score.

---

> > ### Comment · Reviewer_oLSX · 2025-08-04
> >
> > I appreciate the detailed rebuttal. As I mentioned in my initial review, the hierarchical captioning idea is interesting and meaningful. I appreciate the novelty of this paper.  However, the main contribution to the community appears to lie in the dataset aspect, including the sensor-language pair and the captioning. The SenorLM model itself does not feature a specific design tailored for the hierarchical captioning structure. Regarding the scaling law, the analysis is limited to the proposed model and dataset, rather than offering a broader, systematic study with generalizable insights. For these reasons, the work seems more suitable for a dataset track.
> >
> > I appreciate your commitment to open-sourcing the dataset. It would be a good contribution to the community. However, currently it’s difficult to place much weight on such promises, as we have seen many “commitments” that either never materialize or result in disorganized repositories with broken or missing code. More importantly, reviewers currently cannot access or run your code to validate the implementation details. Considering such future commitments as an indicator of reproducibility would be unfair to other submissions that provide well-organized, functioning repositories alongside the paper, allowing reviewers to verify reproducibility directly.

---

> > > ### Author Response · Authors · 2025-08-05
> > >
> > > We appreciate your positive feedback and recognition of our paper’s novelty. **We believe our earlier rebuttal has addressed all other concerns** and would be grateful if the reviewer could update your review and score to reflect them. Below, we clarify the remaining points.
> > >
> > > ### 1. Why this work belongs in the main research track
> > >
> > > **Scope of contribution**. SensorLM tackles the **entire** sensor-language pre-training pipeline, not only data curation. Our work (1) formalizes a scalable procedure for pairing time-series sensor data with language, (2) systematically analyses **what** caption granularity is most effective, (3) establishes empirical scaling laws across **> 30** commercial devices (with OOD test devices) & **60 million** hours of data, and (4) introduces a modular training framework that exposes objective-design trade-offs **not explored** in prior work.
> > >
> > > **Generalizability of insights**. Our scaling experiments draw on a dataset **orders of magnitude larger** than earlier studies (Table 1) and cover heterogeneous wearables from **> 30 device models** (e.g., *Google Pixel Watch 1-3, Fitbit Sense 1-2, Charge 4-6, Versa 1-4, Inspire 2-3*), with separate OOD test devices. This setting reflects **real-world, large-scale deployments**. We welcome any concrete suggestions from the reviewer on how to offer “a broader, systematic study”.
> > >
> > > **Precedent at top venues**. In addition to the novelty highlighted above, we also want to note that empirical studies that advance understanding of wearable foundation models using large datasets with established methods **have been published in the main tracks of leading ML venues**, including **NeurIPS [1], ICLR [2, 3, 4],** and **ICML [5]**. This confirms that work of this nature is squarely in scope.
> > >
> > > References:
> > > 1. UniMTS: Unified Pre-training for Motion Time Series. NeurIPS 2024
> > > 2. Large-scale Training of Foundation Models for Wearable Biosignals. ICLR 2024
> > > 3. Relcon: Relative contrastive learning for a motion foundation model for wearable data. ICLR 2025
> > > 4. Scaling wearable foundation models. ICLR 2025
> > > 5. Beyond Sensor Data: Foundation Models of Behavioral Data from Wearables Improve Health Predictions. ICML 2025
> > >
> > > &nbsp;
> > >
> > > ### 2. Logistics of open-sourcing
> > > We fully support reproducibility and transparency. Because the dataset contains **sensitive health information**, it must undergo rigorous compliance review before release. At submission time, this process was ongoing. We can now confirm that we will release: (1) **a de‑identified version** of the full metabolic downstream dataset, (2) **SensorLM models** pretrained on these data, and (3) **the complete codebase** for caption generation pipeline, pretraining, and evaluation. This release offers a unique, large-scale resource to the community, and we believe our SensorLM framework, models, and unified benchmark will accelerate progress on reproducible, real-world sensor-language foundation models.
> > >
> > > The dataset was collected under IRB-approved protocols with **informed consent** that permits data sharing under appropriate safeguards. Access will be granted to researchers with **verified academic affiliations** who sign a **Data Use Agreement**. Because of these privacy requirements, we are unable to provide a public download link during the anonymized review period.
> > >
> > > &nbsp;
> > >
> > > ---
> > > We believe these clarifications have resolved the remaining concerns adequately. We hope that they will encourage the reviewer to reassess the paper and update their review and score accordingly.

---

> > > > ### Comment · Reviewer_oLSX · 2025-08-06
> > > >
> > > > Thanks for the explanations. I understand that data release needs a rigorous compliance review. However, it is recommended to submit a paper when things are ready, instead of having something ongoing. Otherwise, it is unfair to those authors who submit their papers when everything is ready. At least the authors could submit the code at the initial submission, which is not privacy sensitive. Anyway, I will reconsider my score.

---

> > > > > ### Author Response · Authors · 2025-08-06
> > > > >
> > > > > Thank you for your reply! We are grateful for your feedback and engagement, which has helped us improve our paper. We are glad to hear that our responses have addressed all your concerns and appreciate your willingness to increase your score.

---

### Official Review · Reviewer_wWho · 2025-07-01

**Clarity:** 3
**Significance:** 2
**Originality:** 2
**Rating:** 3
**Confidence:** 4

**Summary:**

This work proposes a way to teach foundation models the “language” of wrist-worn sensors. The authors build a hierarchical captioning pipeline that turns each 24-hour, 26-feature sensor window into statistical, structural and semantic sentences. They train transformer-based dual-encoder + decoder models with combined contrastive-and-generative objectives, unifying CLIP, Cap and CoCa styles in their framework. In zero-shot activity recognition, cross-modal retrieval and few-shot health prediction, SensorLM beats large LLM-tabular baselines and sensor encoders (e.g., DINO, MSN, etc) by sizable margins.

**Questions:**

Please refer to the weaknesses part.

**Ethical Concerns:**

["NO or VERY MINOR ethics concerns only"]

**Final Justification:**

During the rebuttal, the authors have provided new information and new experimental results. However, after considering other reviewers' comments, I would like to keep my score which I think reflects the evaluation of the submitted manuscript.

**Limitations:**

I think some further discussions about fixed feature dependency would be good.

**Paper Formatting Concerns:**

No.

**Quality:**

3

**Strengths And Weaknesses:**

Strengths:

+ The paper delivers a substantial engineering effort for wrist-sensor data. Combined contrastive + generative training unifies CLIP-, caption- and CoCa-style objectives and is backed by ablation and scaling-law studies. Experiments cover zero-shot, few-shot and linear-probe settings and show clear gains over several baselines.
+ The pipeline is illustrated step-by-step; model architecture and objectives are mathematically specified; Other details are moved to a well-structured appendix.

Weaknesses:

- All downstream tasks rely on a fixed 26 features. The framework is hard-coded to those 26 features from a single wrist device. It is unclear how it adapts when tasks demand additional statistics or when multiple sensors each supply their own 26-D vectors.
- Zero-shot evaluation is compared only with Gemini/Gemma (LLM-on-tabular). Strong raw-data time-series foundation models capable of zero-shot representation learning (e.g., Mantis, Moment, Chronos, UniMTS, etc) are omitted, making claims of superiority less convincing.
- Training and test sets come from the same user cohort and device (Section 5), yet the paper emphasises “generalisation”. A leave-person-out (cross-device) split is required to demonstrate genuine robustness.
- The architecture largely reuses established image-text paradigms (ITC, CoCa) with minimal conceptual change.

---

> ### Author Rebuttal · Authors · 2025-07-31
>
> Thank you for your thoughtful and constructive feedback! We are happy to learn that you found the experiments extensive and the paper well-structured. Below, we provide additional results and clarifications, which we hope will encourage the reviewer to increase the score.
>
> &nbsp;
>
> > *W1: All downstream tasks rely on a fixed 26 features... multiple sensors each supply their own 26-D vectors.*
>
> Thank you for raising this. We provide further clarifications and results as follows.
>
> First, we use 26 features (Table 6, Appendix A.1) as a ***generalizable superset*** derived from common smartwatch sensors (ACC, PPG, EDA, etc.) across **> 30** commercial devices. Our pretraining set covers **18** device models (e.g., *Google Pixel Watch 1-3, Fitbit Sense 1-2, Charge 4-6, Versa 1-4, Inspire 2-3*), while the metabolic downstream set has an **additional 17** devices (e.g., *Fitbit Flex, One, Zip, Charge HR, Charge 3*). The superior performance across devices/populations (**Fig. 5, Table 19**) validates the generalizability of derived features, which reflects **real-world** deployment across hardware platforms.
>
> In addition, **not all** devices offer the complete set of 26 features. In both pre‑training and downstream sets, sensor missingness arises naturally (e.g., many devices like *Fitbit Versa* or *Charge* lack EDA sensors). For instance, the Metabolic set has on average 11 ($\pm$ 5) sensor features completely absent; in such cases we imputed values using the population mean computed from pre‑training data, following established practice [ref 6, ref 20].
>
> Consequently, we report comparisons when certain sensor modalities are unavailable:
>
> |Modality Missing|Activity|Anxiety|Hypertension|
> |-|-|-|-|
> |w/o missing|0.94|0.67|0.62|
> |w/ missing|0.89|0.65|0.60|
>
> This confirms that SensorLM remains robust to missing sensor modalities (results in the main paper included all missingness scenarios).
>
> While we acknowledge that the current feature set may not be exhaustive, we would like to emphasize that the captioning pipeline and modeling framework are **not hard-coded** to these 26 features. The framework is designed for **extensibility**: new features can be incorporated by extending the text description without architectural modification. While our feature set already aligns with sensors on **30+** commercial devices, device-specific models could also be trained and their embeddings/predictions can be aggregated at inference time [ref 1, ref 33].
>
> To summarize, we believe the current design is extensive and representative for sensor‑language modeling, and we hope it will serve as a sound basis for future work. We will add the above results and discussions on sensor feature dependency in the revised manuscript.
>
> &nbsp;
>
> > *W2: Zero-shot evaluation is compared only with Gemini/Gemma .. making claims of superiority less convincing.*
>
> Thanks for pointing out these related works. We would like to offer clarifications and results as follows.
>
> First, among the cited methods, Moment [1] and Chronos [2] target zero‑shot *forecasting*, not classification or retrieval. Mantis [3] offers *zero-shot embedding extraction* similar to our SSL baselines. All three require training a classifier for downstream evaluation, placing them in the **few‑shot** rather than actual **zero‑shot** setting (i.e., no tuning or adaptation).
>
> **UniMTS** [ref 36] is the closest approach of comparison. We discussed it in our related work and now we directly compare with it. Specifically, it employs a CLIP [ref 21] objective and treats activity descriptions as text input, corresponding to the *semantic* caption in our framework. We have adapted and trained UniMTS under our setup and report the results below:
>
> | |**Activity (20-class)**|**gym cardio**|**outdoor sports**|
> |-|-|-|-|
> |UniMTS|0.71|0.73|0.65|
> |SensorLM|**0.84**|**0.76**|**0.86**|
>
> The table confirms that SensorLM outperforms UniMTS across zero-shot tasks. We note that the benefits stem from (1) our hierarchical caption design, where relying solely on semantic captions is **sub‑optimal** across tasks (**Table 4, Sec. 5.3**), and (2) our *modularized* training framework, which enables systematic comparison of objectives (CoCa proves more effective than CLIP; **Table 5, Sec. 5.3**).
>
>
> We will incorporate this additional baseline and extend the discussion of related works as suggested by the reviewer in the revised manuscript.
>
> &nbsp;
>
> > *W3: Training and test sets come from the same user cohort ... demonstrate genuine robustness.*
>
> We believe there is a misunderstanding here by the reviewer. We would like to clarify that all tasks in our paper employ a **leave-person-out** evaluation protocol, as specified by the train and test cohorts in **Table 7 (Appendix)**.  Our statement that the data are “from the same population” in Sec. 5 refers only to shared recruitment criteria and overall distribution, not to **any overlap** of participants between training and test sets.  We will revise this wording to eliminate ambiguity and emphasize our leave-person-out evaluation.
>
> On a related note, in addition to the default leave‑person‑out evaluation, we also assess **cross‑device generalization** on the metabolic tasks by comparing **in-domain (N=83,243)** and **out-of-domain (N=6,943)** devices (as discussed in **W1**).  Following the protocol in Sec. 5.1, we report few-shot classification and regression results:
>
> | |Anxiety $\uparrow$|Hypertension $\downarrow$|Age (MAE) $\downarrow$|BMI (MAE) $\downarrow$|
> |-|-|-|-|-|
> |ID|0.65|0.61|9.24|5.74|
> |OOD|0.72|0.55|8.55|5.79|
>
> As shown, the model maintains strong OOD performance: while “BMI” and “Hypertension” show minor variation, “Anxiety” and “Age” exhibit slight improvements under OOD settings.
>
> We believe the clarified leave‑person‑out protocol and the results on OOD devices address the concern and demonstrate SensorLM’s robustness. We will incorporate the analysis and update the manuscript accordingly.
>
> &nbsp;
>
> > *W4: The architecture largely reuses established image-text paradigms (ITC, CoCa) with minimal conceptual change.*
>
> We appreciate the comment and would like to clarify our architectural contribution. Although the architecture skeleton follows established image-text paradigms, our work introduces a **modular** and **generic** pre‑training framework that makes it straightforward to *attach, remove, or re‑weight* objectives. This **modularity** allows systematic ablations, enables rapid integration of new components, and reveals design trade‑offs not explored in prior image-text studies.
>
> To support this claim, we study two extensions that would be less practical without such modular design.
>
> **(1) Loss-weight exploration**. We sweep the relative weights of the contrastive ($\lambda_{con}$) and captioning ($\lambda_{cap}$) losses beyond fixed settings by CLIP ($\lambda_{con}=0$) and CoCa ($\lambda_{con}=\lambda_{cap}=1$). We find clear task‑specific trade‑offs: a 1:2 balance improves zero‑shot activity recognition, whereas 5:1 yields the best metabolic‑health prediction. These results indicate that careful tuning, enabled by our framework, yields measurable gains:
>
> |$\lambda_{con}$|$\lambda_{cap}$|zero-shot|||Linear Probing|||
> |-|-|-|-|-|-|-|-|
> |||**Activity (20-class)**|**gym cardio**|**outdoor sports**|**Activity**|**Anxiety**|**Hypertension**|
> |1|0.2|0.82|0.77|0.82|0.92|**0.66**|**0.62**|
> |1|1|0.84|0.76|**0.86**|**0.94**|0.65|0.60|
> |1|2|**0.85**|**0.78**|0.85|0.93|0.65|0.60|
>
> **(2) Sensor-reconstruction head.**  We append either an auto‑encoder (AE) or vector‑quantized auto‑encoder (VQ‑AE) decoder to the sensor encoder so that training jointly optimizes *contrastive alignment*, *caption generation*, and *signal reconstruction*. Adding the reconstruction loss improves health‑prediction metrics with only a modest drop in activity recognition. The VQ‑AE variant achieves the best overall trade‑off:
>
> | |Zero-Shot|||Linear Probing|||
> |-|-|-|-|-|-|-|
> ||**Activity (20-class)**|**gym cardio**|**outdoor sports**|**Activity**|**Anxiety**|**Hypertension**|
> |Con + Cap|**0.84**|**0.76**|**0.86**|**0.94**|0.65|0.60|
> |Con + Cap + AE|0.76|0.67|0.82|0.89|0.68|0.63|
> |Con + Cap + VQ-AE|0.77|0.68|0.74|0.91|**0.69**|**0.65**|
>
> These straightforward extensions, enabled by the flexibility of the modular framework, demonstrate its support for rapid experimentation and yield concrete performance benefits. This is consistent with the **NeurIPS reviewer guideline**:
>
> > *”originality does not necessarily require introducing an entirely new method when the work enables new insights or capabilities"*
>
> We hope these clarifications and new results highlight the significance of our contribution. We will incorporate the above discussions and results into the manuscript.
>
> ---
> References:
> 1. Goswami et al. "Moment: A family of open time-series foundation models." ICML 2024.
> 2. Ansari et al. "Chronos: Learning the language of time series." TMLR 2024.
> 3. Feofanov et al. "Mantis: Lightweight calibrated foundation model for user-friendly time series classification." arXiv (2025).
>
> &nbsp;
>
> ---
> We hope the above responses have addressed all concerns from the reviewer adequately. We would also like to update the reviewer on our commitment to open‑sourcing SensorLM:
> - **Dataset**. Under our IRB‑approved protocol, we have obtained informed consent to release a **de‑identified version** of the downstream dataset, including hypertension, anxiety, age, and BMI labels. Access will be limited to researchers with verified academic affiliations.
> - **Pretrained model**. We will release **SensorLM checkpoints** trained on these data together with example code for inference and fine-tuning.
> - **Code**. We will open-source the complete caption-generation pipeline to facilitate reuse by the broader research community.
>
> Once again, thanks for your time and constructive feedback. We would really appreciate it if the reviewer could consider raising their scores and championing the paper.

---

> > ### Author Response · Authors · 2025-08-05
> >
> > We would like to thank the reviewer again for your time and effort. We would greatly appreciate the opportunity to engage with you, as we have put a significant effort into providing a detailed response to your questions, including new experiments on **additional baselines**, **OOD generalizability**, and the **design trade-offs** enabled by our general pretraining framework. We believe the additional results and clarifications should have successfully addressed all your concerns.
> >
> > Given there is only **2 days** left in the discussion period, we wanted to double check whether there were any final clarifications the reviewer would like. If the concerns are clarified and the reviewer is convinced of the novelty and completeness of our work, we'd be grateful if the reviewer could update your review and score to reflect that, so we know that our response has been seen. Once again, thanks for your time and dedication to the review process.

---

> ### Comment · Reviewer_wWho · 2025-08-05
>
> I appreciate the effort that the authors put into this rebuttal. In the rebuttal, the authors promised to release the downstream dataset, pretrained model and the codes. It's a pity that the full dataset cannot be released (which I totally understand), as the power of this work is the large-scale captioned sensor data. As demonstrated in the experiments, scaling plays a big role in this work (even though the gains beyond 12 million hours diminish). With such large-scale data, it is possible to leverage a relatively simple model architecture to achieve good downstream performance.

---

> > ### Author Response · Authors · 2025-08-05
> >
> > We appreciate your positive feedback! **We believe our earlier rebuttal has addressed all your concerns**. We’d be grateful if the reviewer could update your review and score to reflect them.
> >
> > We would like to provide further clarification regarding the size of the dataset release. While our consent language prevents us from releasing the entire 59M-hour pretraining dataset, the dataset we are approved to release will include **5.8M person-hours of wearable sensor data (241,532 day-long instances)**. This includes all derived features from the five wearable sensing modalities (i.e. PPG, Accelerometer, EDA, Altimeter, Temperature). As our scaling analysis in **Fig. 8** (Appendix) demonstrates, models trained on this dataset size achieve performance that is already close to the full-scale results. We believe this release, which expands the available data for sensor-language foundation model training by **1,000X** over existing studies (see Table 1), will be an extremely valuable resource for the community for future research.
> >
> > Once again, we thank the reviewer for their time and constructive feedback. We are happy to address any further questions. Given that your latest comments suggest we have addressed **all** your concerns and misunderstandings, we respectfully request that you could update your review and score to reflect them accordingly.

---

### Official Review · Reviewer_yhqJ · 2025-07-10

**Clarity:** 3
**Significance:** 3
**Originality:** 3
**Rating:** 4
**Confidence:** 3

**Summary:**

This work constructs a large-scale multimodal sensory dataset and presents a well-structured sensor-data–caption pretraining pipeline that generalizes effectively across multiple downstream tasks. It directly addresses the challenge of data scarcity, particularly the lack of aligned sensor data, in this domain. The pretrained model demonstrates strong performance across a range of evaluation settings.

**Questions:**

1. Can the proposed learning pipeline or pretrained model generalize to out-of-domain data?

2. Does the modality alignment strategy risk weakening modality-specific representations?

3. Can SensorLM handle scenarios with missing modalities or adapt to varying modality combinations?

4. Could the authors provide more details on the design and implementation of the “hierarchical” caption structure?

5. Is the proposed approach extendable to unaligned multimodal data?

**Ethical Concerns:**

["NO or VERY MINOR ethics concerns only"]

**Final Justification:**

The authors’ rebuttal has addressed my main concerns, and I will maintain my positive rating.

**Limitations:**

yes

**Quality:**

3

**Strengths And Weaknesses:**

Strengths:

+ The scale of this work is large, making a good contribution to the sensory data learning field, which suffers from data scarcity and lack of aligned data.

+ The paper addresses the lack of interpretability of sensory data and forms the captions into structured and interpretable formats. The hierarchical structure is interesting.

+ The evaluation is extensive and convincing compared to many advanced pipelines.

Weaknesses:

- My primary concern is whether the proposed learning pipeline and pretrained model can generalize to out-of-domain data. While the evaluation is extensive, all experiments are conducted on the authors’ collected datasets. In practice, users may employ different device types or encounter variations in data formats. Given the inherent heterogeneity in devices and environments in sensory data applications, it is unclear whether the framework can effectively interpret preprocessed or out-of-distribution data without performance degradation.

- The alignment strategy combining contrastive and generative losses is effective for joint sensor-language modeling. However, enforcing all modalities into a shared latent space may compromise the fidelity of modality-specific representations, especially for time-series sensor data. A more fine-grained approach that applies alignment at different levels based on context could improve performance. In the current design, modality alignment appears uniform and context-agnostic.

- It is unclear whether SensorLM can handle missing modality scenarios or adapt to varying modality combinations. In real-world applications, users may choose not to wear certain sensors due to personal preference, battery constraints, or hardware availability. The model’s flexibility in such cases should be discussed or evaluated.

- The paper briefly mentions a hierarchical design for captions but lacks sufficient detail on its structure. It would be helpful to elaborate on how the statistical, structural, and semantic levels are defined and how the model captures both shared and modality-specific information across these layers.

- Given the large number of participants involved in data collection, more details on the labeling process, both for hard labels and soft labels, would be valuable. For smaller research groups looking to adopt this approach, practical guidance on data annotation would improve reproducibility and usability.

- In many real-world scenarios, fully aligned multimodal sensory data is not available. It would strengthen the work to discuss whether and how the proposed approach can be extended to handle unaligned or weakly aligned sensory inputs.

---

> ### Author Rebuttal · Authors · 2025-07-31
>
> Thank you for your supportive remarks! We are delighted that the experiments and analyses were interesting and thorough to you. Below, we address your concerns in detail. We hope that these will further clarify our work and lead to a favorable increase of the score.
>
> &nbsp;
>
> > *W1 & Q1: Can the proposed pipeline or pretrained model generalize to out-of-domain data?*
>
> Thank you for your constructive comment. We agree that out-of-domain generalization is crucial. First, we would like to clarify that we trained and tested our models with a **diverse set of device types (more than 30)**. Our pretraining set included **18** smartwatch and fitness-tracker models (e.g., *Google Pixel Watch 1-3, Fitbit Sense 1-2, Charge 4-6, Versa 1-4, Inspire 2-3*). The activity downstream dataset used the same devices, while the metabolic downstream dataset has an **additional 17** device types (e.g., *Fitbit Flex, One, Zip, Charge HR, Charge 3*). A complete device distribution will be provided in the Appendix.
>
> As a result, we are able to demonstrate generalizability on the metabolic test set by comparing performance on **in-domain (N=83,243)** and **out-of-domain (N=6,943)** devices. We follow the setting in the main paper (**Sec. 5.1**) and report classification (AUROC) and regression (MAE) results:
>
> | |Anxiety $\uparrow$|Hypertension $\downarrow$|Age $\downarrow$|BMI $\downarrow$|
> |-|-|-|-|-|
> |ID|0.65|0.61|9.24|5.74|
> |OOD|0.72|0.55|8.55|5.79|
>
> As the table confirms, the model exhibits sound OOD generalization: while “BMI” and “Hypertension” predictions show slight variation, “Anxiety” and “Age” predictions may improve under OOD conditions.
>
> Moreover, although training a foundation model on every possible device is impractical, our approach mitigates this challenge by operating on **minutely features** (e.g., heart rate, step count) derived from raw sensor data, which are inherently more consistent across devices. To support this, we present the distribution of “heart rate” measurements across representative devices:
>
> |Device|HR|
> |-|-|
> |Pixel Watch 2|79.6$\pm$3.9|
> |Charge 6|78.2$\pm$3.8|
> |Sense 2|79.3$\pm$3.6|
> |Versa 4|78.5$\pm$3.7|
> |Inspire 3|78.2$\pm$3.6|
>
> This confirms the robustness of the derived features across hardwares. We will add the results to the revised manuscript.
>
> &nbsp;
>
> > *W2 & Q2: Does the modality alignment strategy weaken modality-specific representations?*
>
> Thanks for the question. We provide further discussions and clarifications as follows.
>
> Existing work in the image-language domain indicates that straightforward vision-language alignment objectives, when trained on large datasets, yield vision‑only encoders with SOTA performance [ref 21, ref 33]. Following the empirical scaling behavior observed for sensor-text data (see **Sec. 5.2**), we expect that the same simple objective, applied at scale, will produce strong sensor‑only encoders.
>
> Our experiments support this: **Fig. 5** and **Appendix C.2** compare embeddings learned by our sensor-language model with sensor-only baselines, showing that our model outperforms across tasks and data sizes. This suggests that the alignment strategy preserves modality‑specific information relative to unimodal objectives.
>
> To summarize, we believe the current design should be extensive and representative for sensor-language modeling research. We hope it will serve as a sound basis and inspire future work. We certainly welcome any further specific suggestions regarding possible extensions.
>
> &nbsp;
>
> > *W3 & Q3: Can SensorLM handle scenarios with missing modalities or varying modality combinations?*
>
> Thanks for pointing this out, and we apologize if it wasn’t clear. In fact, in both our pretraining and downstream datasets, sensor missingness arises naturally (e.g., devices like *Fitbit Versa* or *Charge* lack EDA sensors). We have performed thorough preprocessing for data missingness following established practices [ref 6, ref 20]:
> When an entire sensor modality is absent, we impute values with the population mean computed from the pretraining data.
> For temporal gaps within a sensor stream (e.g., dropouts), we apply linear interpolation across gaps and forward/backward filling at sequence boundaries.
>
> Consequently, SensorLM is trained on data with up to **80%** missingness. It learns to model missingness patterns and shows robustness across scenarios.
>
> To further address the reviewer’s concern about performance under missingness, we conducted additional experiments on (1) subgroup analysis stratified by percentage of missingness, and (2) comparisons when certain sensor modalities are entirely absent.
>
> **First**, we observe only mild degradation as missingness increases from <40% to 60–80%:
>
> |Missing %|Activity|Anxiety|Hypertension|
> |-|-|-|-|
> |<40%|0.94|0.68|0.62|
> |40-60%|0.94|0.65|0.59|
> |60-80%|0.91|0.65|0.65|
>
> **Second**, when specific sensor modalities (e.g., EDA or HRV) are entirely absent, we note a moderate AUROC decrease (0.02–0.09) relative to the full‑modality setting:
>
> |Modality Missing|Activity|Anxiety|Hypertension|
> |-|-|-|-|
> |w/o missing|0.94|0.67|0.62|
> |w/ missing|0.89|0.65|0.60|
>
> These findings confirm that, although performance degrades slightly under missingness, SensorLM remains robust and flexible with incomplete sensor data. All results reported in the paper are aggregated over the entire test set, which already includes these missing‑data scenarios. We will incorporate the new analyses in the revised manuscript.
>
>
> &nbsp;
>
> > *W4 & Q4: Could the authors provide more details on the design and implementation of the hierarchical caption?*
>
> We appreciate the reviewer’s interest. We would like to clarify that definitions of the statistical, structural, and semantic levels are presented in **Fig. 1B** and **Fig. 3**, with rewrite templates in **Appendix A.2**. We provided additional representative examples in **Fig. 4** and **Figs. 10, 11**.
>
> When levels are combined, captions are concatenated in the sequence: (semantic, structural, statistical). Their interaction and individual contributions are studied in **Table 4** and **Table 23 (Appendix)**, where results indicate that leveraging both shared and modality-specific information improves downstream performance.
>
> For completeness, we will include a full example in the Appendix. We welcome any further questions regarding specific elements of the caption hierarchy.
>
>
> &nbsp;
>
> > *W5: Given the large number of participants involved … practical guidance on data annotation would improve reproducibility and usability.*
>
> Thank you for the suggestion. We would like to refer the reviewer to details in **Sec. 3.2** and **Sec. 5**.
>
> For **semantic** captions and downstream task labels, all **event-level** information (including user activity, mood, disease diagnoses, demographic data) is *self-reported* by the user. This constitutes our "hard" labeling. Sensor-derived **statistics** and **structural** patterns are computed ***programmatically*** from raw signals. We apply sliding windows and fit linear regression within each window to categorize trends (e.g., increasing, decreasing) based on slope thresholds. We also use peak detection algorithms (`scipy.signal.find_peaks`) to identify spikes and drops.
>
> This design ensures the approach is both **scalable** and **reproducible**. Structural and statistical captions can be derived from any comparable sensor data, and the semantic component can be extended to other self-reported or externally annotated information.
>
> We fully agree with the reviewer that this pipeline can benefit the broader research community. To further support adoptions, we will open source the captioning code and provide practical documentation in the final release.
>
>
> &nbsp;
>
> > *W6 & Q5: Is the proposed approach extendable to unaligned multimodal data?*
>
> Thanks for raising this. We would like to further discuss several aspects.
>
> First, for the input features, we compute minutely features from raw sensor signals, which reduces misalignment issues when aggregating data from diverse devices and sensors [ref 6, ref 20]. The OOD performance reported in **W1 & Q1** supports its robustness.
>
> In addition, when data from certain sensor channels or intervals are missing, we apply imputation and related preprocessing procedures established in prior work. The results reported in **W3 & Q3** confirms that SensorLM generalizes well under these conditions.
>
> Beyond these measures, SensorLM is readily extendable to handle more irregular data. This can be achieved by integrating established methods (e.g., appending a missingness mask or using alternative tokenization strategies [1, 2]) within the sensor encoder, which are orthogonal to our main framework and can be seamlessly incorporated.
>
> We will add the above analyses and discussions in the revised manuscript.
>
> ---
> References:
> 1. Beebe-Wang et al. "PAITS: pretraining and augmentation for irregularly-sampled time series." arXiv preprint (2023).
> 2. Erturk et al. "Beyond Sensor Data: Foundation Models of Behavioral Data from Wearables Improve Health Predictions." ICML 2025.
>
> &nbsp;
>
> ---
> We hope the above responses have addressed all concerns from the reviewer adequately. We would also like to update the reviewer on our commitment to open‑sourcing SensorLM:
> - **Dataset**. Under our IRB‑approved protocol, we have obtained informed consent to release a **de‑identified version** of the downstream dataset, including hypertension, anxiety, age, and BMI labels. Access will be limited to researchers with verified academic affiliations.
> - **Pretrained model**. We will release **SensorLM checkpoints** trained on these data together with example code for inference and fine-tuning.
> - **Code**. We will open-source the complete caption-generation pipeline to facilitate reuse by the broader research community.
>
> Once again, thanks for your time and constructive feedback. We would really appreciate it if the reviewer could consider raising their scores and championing the paper.

---

> > ### Author Response · Authors · 2025-08-05
> >
> > We would like to thank the reviewer again for your time and effort. We would greatly appreciate the opportunity to engage with you, as we have put a significant effort into providing a detailed response to your questions, including new experiments on **OOD generalizability** and **robustness to missing modalities**. We believe the additional results and clarifications should have successfully addressed all your concerns.
> >
> > Given there is only **2 days** left in the discussion period, we wanted to double check whether there were any final clarifications the reviewer would like. If the concerns are clarified and the reviewer is convinced of the novelty and completeness of our work, we'd be grateful if the reviewer could update your review and score to reflect that, so we know that our response has been seen. Once again, thanks for your time and dedication to the review process.

---

> > ### Comment · Reviewer_yhqJ · 2025-08-06
> > **Official Comment by Reviewer yhqJ**
> >
> > Thanks for the authors’ rebuttal, which addressed my main concerns.

---

> > > ### Author Response · Authors · 2025-08-06
> > >
> > > Thank you for your positive remark! We are glad that our responses have addressed your concerns. We are happy to address any further questions. Given all concerns have been clarified, we would be grateful if the reviewer could consider updating your review and score to reflect them.

---

### Note · Authors · 2025-08-12

We sincerely thank all reviewers and ACs for your time and dedication throughout the review process, which have helped improve our paper. We are delighted that the reviewers highlighted the scale of the work, the hierarchical captioning approach, and the comprehensive evaluations as clear strengths. We are also glad that our rebuttal successfully resolved concerns about novelty, generalization, and robustness. As a result, we are encouraged to see that we have addressed **all concerns** raised by **all reviewers**, and we appreciate the strong positive reception of SensorLM.

We remain committed to **open-sourcing** our complete codebase, the large-scale de-identified dataset, and our pretrained models to support future research in the community. We believe these contributions and commitments reflect both the technical rigor and the practical impact of our work.

We appreciate the emerging consensus among reviewers recognizing SensorLM’s contributions. Again, we are grateful for the active and constructive discussions and hope the final recommendation will be based on a holistic assessment of the manuscript and aligned with the positive outcome of these discussions.

---

### Decision · Program_Chairs · 2025-09-17

**Decision:**

Accept (poster)

**Comment:**

Paper present a large-scale dataset (of subjects and different devices, up to date the largest collected dataset in this domain), sensor data captioning, and pre-training framework for sensor-language foundation models for understanding wearable sensor data with natural language outputs. Hierarchical captioning is designed to capture statistical, structural, and semantic information from sensor data. Framework utilises and extends existing pre-training architectures to  unified framework. Framework and proposed dataset are experimentally evaluated in human activity recognition and healthcare applications, and compared to generic multimodal LLMs with promising results.

Proposed work presents timely, interesting, and novel application of multimodal generative models for sensor-language data. Several
strengths are observed. The paper is considered well-written and organised (Reviewer oLSX), and the framework is well-illustrated and defined (Reviewer wWho). Furthermore, introducing  of very large-scale dataset collection and utilisation (Reviewers yhqJ and oLSX) is considered a good contribution and valuable for the field. The evaluations are well-done, extensive, and strong (Reviewers yhqJ, wWho, and ZU6J), and the support of test-time inference is seen valuable and practical (Reviewer ZU6J).

Several weaknesses have been indicated during the review phase, including the questions of model's out-of-domain generalisation capabilities (Reviewer yhqJ), handling of missing modalities (Reviewer yhqJ), limitations of their alignment strategies (Reviewer yhqJ), originality of the model components and loss function approaches (Reviewer oLSX), computational overhead (Reviewer ZU6J), and generalisation of the model to other time-series sensor data applications (Reviewer ZU6J). From evaluation perspectives, limited baseline comparison is indicated (Reviewer oLSX), and from dataset and code perspective, missing details of data processing pipeline (Reviewer yhqJ) as well as missing dataset and code open-sourcing during the review period (Reviewer wWho, oLSX).

During the rebuttal period discussion have been quite active, and authors have clarified the raised issues. Specifically, comprehensive set of new results and clarifications have been produced, which support the claims and addresses the main concerns of the reviewers. What is left is the concerns about data and code open-sourcing for the community. Paper has mixed rating both side of the borderline (from borderline reject to accept), indicating the concerns about data and code publishing and application orientation of the work, in general. Overall, paper has its merits and if all the improvements (including promised data and code releases) are done in the final revision, paper can be recommended for acceptance. Authors are especially encouraged to revise the paper with all promised modifications introduced during the rebuttal phase.